# Abstraction Induces the Brain Alignment of Language and Speech Models

**Emily Cheng** [1]   **Aditya R. Vaidya** [2]   **Richard J. Antonello** [3]

## Abstract

Research has repeatedly demonstrated that intermediate hidden states extracted from large language models and speech audio models predict measured brain response to natural language stimuli. Yet, very little is known about the representation properties that enable this high prediction performance. Why is it the intermediate layers, and not the output layers, that are most effective for this unique and highly general transfer task? We give evidence that the correspondence between speech and language models and the brain derives from shared meaning abstraction and not their next-word prediction properties. In particular, models construct higher-order linguistic features in their middle layers, cued by a peak in the layerwise *intrinsic dimension*, a measure of feature complexity. We show that a layer's intrinsic dimension strongly predicts how well it explains fMRI and ECoG signals; that the relation between intrinsic dimension and brain predictivity arises over model pre-training; and finetuning models to better predict the brain causally increases both representations' intrinsic dimension and their semantic content. Results suggest that semantic richness, high intrinsic dimension, and brain predictivity mirror each other, and that the key driver of model-brain similarity is *rich meaning abstraction* of the inputs, where language modeling is a task complex enough (but perhaps not the only) to require it.

 chengemily1/brain-id-abstract

## 1. Introduction

How do brains and machines take low-level information, such as a collection of sounds or words, and compose it into the rich tapestry of ideas and concepts that can be expressed in natural language? This question of meaning abstraction is at the heart of most studies of human language comprehension. Recent work has shown that representations from contemporary language models (LMs) and speech models are able to successfully model human brain activity at varying spatial and temporal resolutions with only a linear transformation (Goldstein et al., 2022; Vaidya et al., 2022; Jain et al., 2023; Antonello et al., 2023; Tuckute et al., 2024; Oota et al., 2023; Mischler et al., 2024). This has led to questions about the reason for this brain-model similarity. Do models and brains possess similar representations because they have similar learning properties or objectives? (Caucheteux et al., 2023; Schrimpf et al., 2021; Goldstein et al., 2022) Or is the similarity merely a consequence of shared abstraction, the ability to represent features not derivable from the surface properties of language alone? (Antonello & Huth, 2022)

In this work, we present new evidence that it is the abstractive properties of LLMs and speech models that drive predictivity between models and brains. We do this by examining an underexplored and unexplained phenomenon of the similarity—the tendency for intermediate hidden layers of language and speech models to be optimal for this linear transfer task. We show that a hidden layer's performance at predicting brain activity is strongly related to *intrinsic dimension* of that layer relative to other layers in the same network. Prior work has found a peak in layerwise intrinsic dimension of LLMs to mark a phase of abstract linguistic feature building (Cheng et al., 2025), which we confirm for language and speech models; in the latter half of LLMs, these features are gradually refined towards predicting the next token (Lad et al., 2025; Skean et al., 2025). We suggest that it is *meaning abstraction* at the dimensionality peak, rather than prediction, that primarily drives the observed correspondence between brains and language-audio models. This hypothesis is supported with three pieces of evidence: (1) layerwise processing, where the abstractive $I_d$-peak layers are also the ones that best predict the brain; (2) language model pre-training, where intrinsic dimension and encoding performance grow in tandem with linguistic abilities; and (3) finetuning models directly to predict the brain, which increases both the intrinsic dimension and semantic content of representations. Results point to an intricate relationship between a linguistic representation's semantic richness, its

---

[1]Universitat Pompeu Fabra, Barcelona, Spain [2]The University of Texas at Austin, USA [3]Zuckerman Mind Brain Behavior Institute, Columbia University, USA. Correspondence to: Emily Cheng <emily.shanacheng@upf.edu>.

*Proceedings of the 43rd International Conference on Machine Learning*, Seoul, South Korea. PMLR 306, 2026. Copyright 2026 by the author(s).

geometric properties, and its similarity to the brain.

## 2. Related Work

**Encoding models of the brain** Representational alignment between the brain and modern foundation models for language and audio has been well established. Early language encoding models were designed as linear probes that mapped the representations from simple lexical word embedding models such as GloVe and English1000 (Huth et al., 2016) to activations in the brain. Later work demonstrated that contextual language models (Jain & Huth, 2018) and audio models (Vaidya et al., 2022; Millet et al., 2022) were more effective at predicting brain response. Antonello et al. showed that encoding model performance, like many other machine learning tasks, scales with the size of the underlying language or speech model. Recent work (Vattikonda et al., 2025; Moussa et al., 2025) demonstrated that it is even possible to finetune the nonlinear weights of audio models directly on brain responses. These "brain-tuned" models have been shown to have representations that align more to higher-level regions in the brain outside of auditory cortex when compared to their pretrained alternatives.

**Reasons for model-brain alignment** While brain-model alignment across varied modalities is a well-established phenomenon, the underlying reasons for this relationship remain unclear. Some works (Schrimpf et al., 2018; Caucheteux et al., 2023; Goldstein et al., 2021) have suggested that the reason that brains and models are similar is because they share a *predictive coding* learning objective, that is, that brains and models both learn by updating predictions via error signals, and so their representations are similar due to this similar objective. Other works (Antonello & Huth, 2022; Schönmann et al., 2025) have claimed that model-brain alignment is merely a side-effect of the highly general representations that models and brains learn from exposure to similar naturalistic input statistics. In this way, any sufficiently capable model trained on rich real-world data will necessarily capture structure that is present in neural responses, even if the mechanistic implementation of these models differs substantially from biological computation. It remains an open question whether alignment reflects a shared predictive objective or simply the convergence of complex learning systems exposed to the same naturalistic structure (Huh et al., 2024).

**Dimensionality and meaning abstraction in neural networks** To understand how contemporary deep neural networks process their inputs, several works have studied how the geometry of representations evolve over network layers (Ansuini et al., 2019; Valeriani et al., 2023; Lee et al., 2025a). Across vision, language, and protein models, the representations' intrinsic dimension ($I_d$) over layers is characterized by a central high-dimensional peak (Ansuini et al., 2019;

Valeriani et al., 2023; Cheng et al., 2025), where the $I_d$ is computed on a general slice of the model's in-distribution data (e.g., The Pile for LLMs). In the language domain, Cheng et al. (2025) showed that LLMs' $I_d$-peak layers coincide with a locus of rich and abstract feature building; Baroni et al. (2026) showed the $I_d$ peak to demarcate when LLMs start to distinguish between inputs with near-identical surface properties but different syntactic structure. These results corroborate other studies showing that language model processing can be broken down into several stages (Tenney et al., 2019; Jawahar et al., 2019), where models first build up complex features of inputs by the intermediate layers, then resolve towards a next-token distribution (Skean et al., 2025; Lad et al., 2025; Acevedo et al., 2025). Overall, the $I_d$ of representations has shown itself to be a useful indicator of when (over training) and where (over layers) LLMs construct rich, higher-order linguistic features (Chen et al., 2024; Lee et al., 2025b). We extend this finding to speech audio models, which are also known to process their inputs along an acoustic to semantic linguistic hierarchy (Pasad et al., 2021; 2023; He et al., 2025).

Dimensionality of neural network representations has been proposed to partially explain brain predictivity (Canatar et al., 2023; Schaeffer et al., 2024). Vision models better predict brain responses to visual stimuli when their representations of natural images are higher dimensional (Elmoznino & Bonner, 2024); a similar but weaker correlation has been found in audio models (Tuckute et al., 2023). Our work builds on this foundation, but differs in that (1) we focus on a different dimensionality measure, $I_d$, which better predicted brain similarity compared to the linear effective dimension used in cited works; (2) we consider the $I_d$ measured on generic, in-distribution data (Elmoznino & Bonner, 2024), such that it describes the general layer function of the *model*, instead of on specific stimuli given to subjects (Tuckute et al., 2023; Canatar et al., 2023; Schaeffer et al., 2024); (3) we interpret $I_d$ measured on generic linguistic stimuli as a proxy for learned linguistic structure (Cheng et al., 2025) over a single model's layers, rather than as a reason for high predictivity in-and-of itself (Schaeffer et al., 2024).

## 3. Data and Methods

### 3.1. Neural response data

**fMRI** We used data from three subjects, UTS01, UTS02, and UTS03, from LeBel et al. (2023)'s open functional magnetic resonance imaging (fMRI) dataset, where subjects listened to English podcast stories. For encoding model training, each subject listened to approximately 20 hours of data. For model testing, the subjects listened to three test stories (one of them 10 times, 2 of them 5 times each). These responses were then averaged across repetitions. Training and test stimuli are listed in Appendix B. Only voxels within

8 mm of the mid-cortical surface were analyzed, yielding roughly 90,000 voxels per subject. See LeBel et al. (2023) for more details.

**ECoG** Electrocorticography data were sourced from the open-source "Podcast" dataset (Zada et al., 2025), which consists of a single 30 minute podcast collected across 9 subjects. The signal was first re-referenced with common average referencing. A notch filter was then used to remove line noise at 60Hz and its harmonics. Finally, a Butterworth filter was used to extract the high-gamma frequency band from 70Hz to 200Hz. Bad channels were excluded from the data via an analysis of their power spectrum. Full details can be found in the original paper (Zada et al., 2025).

### 3.2. Methods

We test the hypothesis that feature richness is the primary driver of brain-model similarity. To do so requires several observables. First, we measure the dependent variable, **(1)** model-to-brain encoding performance, by scoring the predictions of a learned affine map from LLM or speech model representations to brain activity. Then, we compute the **(2)** $I_d$ of representations to measure feature complexity over the LLM or speech model's layers. **(3)** To link $I_d$ to the linguistic features expressed at each layer, we conduct layerwise probing experiments. Finally, to test the alternate hypothesis that next-token prediction drives brain-model similarity (Schrimpf et al., 2018; Caucheteux et al., 2023; Goldstein et al., 2022), we compute the **(4)** *surprisal*, or next-token prediction error, from each layer. In all cases, we use the last-token residual stream representation, as it is the only one to attend to the entire sequence.

**Encoding performance** To train **fMRI** encoding models, we use the method described in Antonello et al. (2023). For LLM-based encoding models, the procedure is as follows. For each word in the stimulus set, activations were extracted by feeding that word and its preceding context into the LLM. A sliding window was used to ensure each word received a minimum of 256 tokens of context. Activations were then downsampled using a Lanczos filter and FIR delays of 1,2,3 and 4 TRs were added to account for the hemodynamic lag in the BOLD signal. For speech models, we segment the audio stimuli into 16 s chunks with a stride of 100 ms, feed the chunks into the model, and extract the last token representation. We then similarly downsample the representations using the Lanczos filter. Finally, we train a ridge regression from the downsampled, time-delayed features. **ECoG** encoding models are trained similarly, via linear models to predict the electrodes' high-gamma response. For 128 evenly spaced lags between -2 and +2 seconds, a separate model is trained to predict the response offset by that lag from word onset. The encoding performance for a model is chosen as the best predicted lag among this set of 128 lags.

In all cases, encoding performance is evaluated using the Pearson correlation coefficient $R$ on the validation set.

**Dimensionality of representational manifolds** A key ingredient in our analysis is the feature richness at LLM and speech layers. We operationalize feature richness by the layers' representational *dimensionality*, which measures the number of (nonlinear) axes that scaffold the feature space. In particular, since LM representations tend to lie near *nonlinear* manifolds with dimensionality much lower than the ambient dimension (Cai et al., 2021; Cheng et al., 2023), we focus on the nonlinear *intrinsic dimension* $I_d$ of the manifold, though we also tested linear estimators which yielded weaker results, see Appendix C.

We estimated the $I_d$ of activations at each layer using GRIDE (Denti et al., 2022), a state-of-the-art $I_d$ estimator (details in Appendix C). We are interested in an LLM or speech model's behavior on a representative sample of its training data, so that the computed dimensionality is informative about the layer's role in general linguistic processing. For LLMs, we compute the $I_d$ on the layerwise representations of $N = 10000$ random 20-word contexts from Pythia's training data,[1] The Pile (Gao et al., 2020). For speech models, we compute the $I_d$ on $N = 10000$ random audio chunks of at most 20 seconds from LibriSpeech (Panayotov et al., 2015), an audiobook dataset WavLM and Whisper are trained on. We repeat this for 5 bootstraps and report the average $I_d$ over splits.

**Layerwise probing** To ascertain what kind of linguistic information (surface-level vs. higher order) each layer contains, we conduct layerwise linear probing experiments using Conneau et al. (2018)'s SentEval dataset for LLMs, and the acoustic and semantic features in Vattikonda et al. (2025) for speech models. We evaluate whether a certain feature is more or less represented at a layer by the probing test performance (accuracy for SentEval, $R^2$ for speech). For task and implementation details, see Appendix D.

**Surprisal** To test whether predictive coding explains model-brain predictivity in the intermediate layers, we computed the next token's surprisal from the models' intermediate layers via an affine mapping to the vocabulary space (Belrose et al., 2023). We used The Pile for LLMs and LibriSpeech for speech models. See Appendix E for details.

### 3.3. Deep language and speech models

**Language** We use six mid-sized LMs: OPT (125m, 1.3b, and 13b) (Zhang et al., 2022) and Pythia (160m, 410m, and 6.9b) (Biderman et al., 2023), chosen due to the wide range of sizes and training checkpoints for analysis of training dynamics. All are causal, Transformer-based language models trained on large-scale text corpora to minimize the *surprisal*,

---

[1] The training data for OPT are not publicly downloadable.

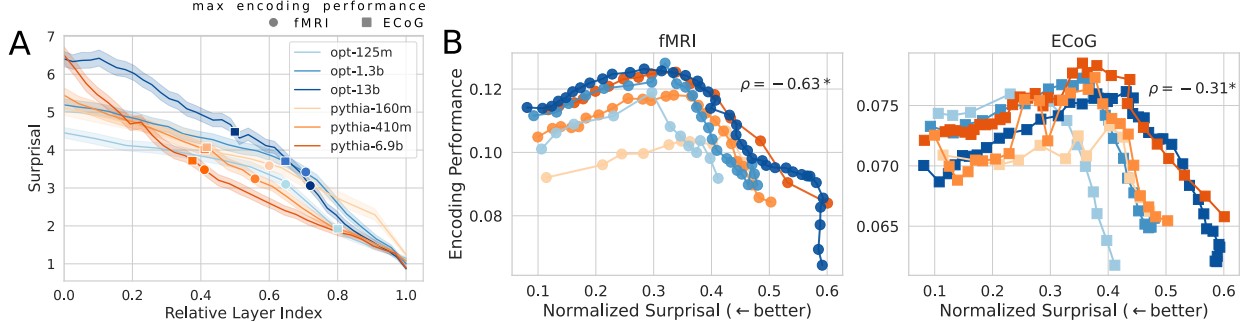

*Figure 1.* **LLM surprisal and encoding performance. (A) Surprisal over LLM layers.** For each LLM, the next-token surprisal from a given layer (y-axis), fit with an affine map (Belrose et al., 2023), decreases monotonically over processing (x-axis). The lowest surprisal layer is always the last one, while the layer most predictive of the brain (○) occurs between 40-70% of processing. **(B) Layerwise encoding performance vs. layerwise surprisal.** For each LLM, processing trajectories through the (encoding performance, surprisal) plane are shown, where surprisals are normalized by the log-vocabulary size for comparability between models. The global Spearman correlation across models is $\rho = -0.63$ for fMRI and $\rho = -0.31$ for ECoG, significant at a $p$-value cutoff of 0.01 (*).

or negative log-likelihood, of the next token given context.

**Speech** We use three state-of-the-art audio models, WavLM (base-plus and large) (Chen et al., 2022) and Whisper (large) (Radford et al., 2023). WavLM is a Transformer-based model trained on masked speech prediction and denoising, and Whisper is an encoder-decoder model trained on automatic speech recognition, both on large-scale speech corpora. In this work, we only consider Whisper's encoder, as it serves as the model's speech featurizer.

**Random baseline** As a baseline, we generate Random Fourier Features (RFF) (Rahimi & Recht, 2007) to align both fMRI and ECoG datasets to a shared, model-agnostic representation space that has not learned to efficiently embed the structure of language. We construct RFF maps of increasing extrinsic dimension (128, 256, 512, 1024, 2048) by drawing a Gaussian random projection and applying sinusoidal nonlinearities to approximate an RBF kernel. For each word we assign a random input vector, transform it through the feature map, then resample to the appropriate frequency according to whether we predict fMRI or ECoG.

## 4. Results

Across language and speech models, subjects, and imaging modalities, experiments evidenced a strong relationship between feature richness ($I_d$), linguistic meaning abstraction, and encoding performance. Here, we report ECoG averaged over all 9 subjects and fMRI averaged over all three subjects, with individual fMRI subjects in Appendix H.

### 4.1. Next-token predictivity does not account for layerwise encoding performance

Several studies (Schrimpf et al., 2021; Caucheteux & King, 2022) have found language model surprisal to predict language encoding performance. We first asked whether sur-

prisal (1) predicts the best performing layer, and more generally, (2) accounts for layerwise variation in encoding performance, finding neither hypothesis to hold across models.

The best layer for next-token prediction does not predict the best encoding performance layer. Figure 1A shows for LLMs that next-token predictivity monotonically decreases over layers (see Figure E.1 for speech models), where the most predictive layer is always the *last* one; instead, encoding performance peaks for some *intermediate* layer (solid dots in Figure 1A). Nor does layerwise surprisal account for layerwise variation in encoding performance: across models, Figure 1B plots each LLM layer's surprisal against its encoding performance averaged over fMRI and ECoG subjects (see Figure E.1 for speech models). If layerwise surprisal predicted layerwise encoding performance, the Spearman $\rho$ should be near $-1$; instead, Table 1 (right) shows that across all models, the global $\rho$ between surprisal and encoding performance is $-0.53$ (fMRI) and $0.21$ (ECoG). *Within* each model, Table 1 rows 1 and 3 show the layerwise $\rho$ between surprisal and encoding performance. Results varied widely between models: $\rho$ was negative for OPT and Whisper ($p < 0.05$), often not significant for Pythia, and sometimes positive for WavLM. Results held for each subject; see Table H.1.

Separately for fMRI and ECoG, we fit a mixed-effects model on $z$-scored layerwise quantities: mean encoding performance over subjects $\sim$ surprisal + $I_d$, with group effects on the model. On fMRI, $I_d$ ($p$-value $\approx 0$) explained away the effect of surprisal ($p$-value $= 0.99$). The effect size for $I_d$ was $\beta_{I_d} = 0.63$, that is, each standard deviation increase in $I_d$ tracks a 0.63 standard deviation increase in encoding performance (in contrast, $\beta_{\text{surprisal}} \approx 0$). For ECoG, both regressors were significant at $\alpha = 0.01$, but the effect size $\beta_{I_d} = 0.97$ for $I_d$ was roughly four times higher than $\beta_{\text{surprisal}} = 0.26$. We conclude that in the general case,

*Table 1.* For fMRI and ECoG (averaged over participants), the Spearman correlations between layerwise surprisal and layerwise encoding performance averaged over voxels/electrodes is in the top row, and correlations between $I_d$ and encoding performance in the bottom row. In all models, Spearman $\rho$ between surprisal and encoding performance varies by model family (negative is better), while $\rho$ between $I_d$ and encoding performance is significantly positive (higher is better). The positive correlation magnitude between $I_d$ and encoding performance is almost always **higher** than or as high as the negative correlation magnitude between surprisal and encoding performance, showing that $I_d$ better explains layerwise encoding performance than surprisal. Values significant at $\alpha = 0.05$, as determined by a permutation test, are marked with *.

| | | OPT | | | Pythia | | | WavLM | | Whisper | LLMs | Speech | All |
|---|---|---|---|---|---|---|---|---|---|---|---|---|---|
| | | 125m | 1.3b | 13b | 160m | 410m | 6.9b | base | large | large | | | |
| fMRI | Surprisal ↓ | -0.62* | -0.82* | -0.83* | 0.33 | -0.39 | -0.27 | 0.85* | 0.83* | **-0.98*** | -0.63* | 0.56* | -0.53* |
| | $I_d$ ↑ | **0.90*** | **0.82*** | **0.85*** | **0.33** | **0.91*** | **0.94*** | -0.14 | **0.90*** | 0.97* | **0.88*** | **0.40*** | **0.72*** |
| ECoG | Surprisal ↓ | -0.15 | -0.41* | -0.53* | -0.61* | -0.60* | 0.33 | -0.09 | 0.60* | **-0.64*** | -0.31* | 0.10 | 0.21* |
| | $I_d$ ↑ | **0.97*** | **0.97*** | **0.82*** | **0.83*** | **0.88*** | **0.88*** | **0.21** | **0.42*** | **0.64*** | **0.83*** | **0.29*** | **0.43*** |

layerwise $I_d$ is a better candidate than layerwise surprisal when accounting for both encoding performance and the best performing layer; this rules out next-token predictivity as a primary driver of layerwise encoding performance.

## 4.2. Linguistic feature abstraction drives encoding performance

We now consider the alternate hypothesis: deep speech and language model layers predict neural responses to language because LMs and speech models learn a rich feature space of language (Antonello & Huth, 2022). The link between high $I_d$ at a layer and its role in constructing syntactic and semantic features has been demonstrated in LLMs (Cheng et al., 2025; Lee et al., 2025b; Baroni et al., 2026), and here we also show this applies to speech models. If our hypothesis holds, then we would expect to see a positive relationship between layerwise $I_d$ and layerwise encoding performance. In what follows, we indeed find a **strong correspondence between $I_d$ and encoding performance** as evidenced by the evolution of $I_d$ and encoding performance over layers, over language model pre-training, and after finetuning model layers directly on neural responses.

$I_d$ **tracks linguistic abstraction in the deep layers of LLMs and speech models**   We first confirm the link between an $I_d$ peak and the construction of higher-order linguistic features in the deep layers of LLMs and speech models. Figure 2A shows, for LLMs (top left) and speech models (bottom left), that the $I_d$ tends to peak in the middle layers. Then, Figure 3 shows that for LLMs (top two rows), semantic decodability (purple) reaches a maximum around the $I_d$-peak layers, while surface feature decodability (pink) does not relate to the $I_d$ peak. Similarly, in speech audio models (bottom row), acoustics are more linearly represented in the early layers, while semantics are most decodable in late layers coinciding with high $I_d$.

$I_d$ **accounts for layerwise brain encoding performance** Figure 2A shows the evolution of $I_d$ over the layers for LLMs and speech models (left) and the layerwise encoding performance averaged over fMRI (middle) and ECoG sub-

jects (right). There is a striking resemblance between layerwise $I_d$ and encoding performance for all models, subjects, and imaging modalities. Notably, for every combination, the $I_d$-peak layers are the ones that best predict neural responses to language, where over all models, the worst case distance between the max-$I_d$ and max-encoding performance layer was only 5 layers (out of total 40 layers, OPT-13b), and usually 0-1 layers (see Table G.1).

Furthermore, layerwise $I_d$ correlates highly to encoding performance globally across the models—$\rho = 0.72$ for fMRI and $\rho = 0.43$ for ECoG ($p < 1e$-2)—where $I_d$ here is normalized by the $\log$-hidden dimension for comparability across models, see Figure 2B. Within each model, the layerwise $I_d$ correlates highly to layerwise encoding performance in all cases, where the correlation is notably at least as good as the alternative hypothesis (surprisal), see Table 1 for fMRI and ECoG averaged over subjects, and Table H.1 for similar takeaways per fMRI subject. This is true for virtually every combination of language or speech model, imaging modality, and subject. A lone exception is WavLM-base-plus, whose encoding performance we found to be driven by low-level similarity to the auditory cortex, see Figure J.3; this can occur in speech models in particular (Oota et al., 2024). Still, results overwhelmingly show the $I_d$ peak in language-audio models to cue a phase of rich linguistic representation that best predicts brain responses to speech.

**Intrinsic dimension and encoding performance grow in tandem over pre-training**   The relationship between encoding performance and $I_d$ arises nontrivially from learning. Figure 2 (bottom) plots Pythia 6.9b's $I_d$ (**C**), encoding performance for fMRI (**D**) and for ECoG (**E**), across layers over the course of training (takeaways per fMRI subject look similar, see Figure H.2).

Figure 2 replicates two results from the literature: first, the $I_d$ peak emerges and $I_d$ generally grows for all layers over training (Cheng et al., 2025); second, ECoG and fMRI encoding performance grow over training (AlKhamissi et al., 2025). Now considering the relationship between the two,

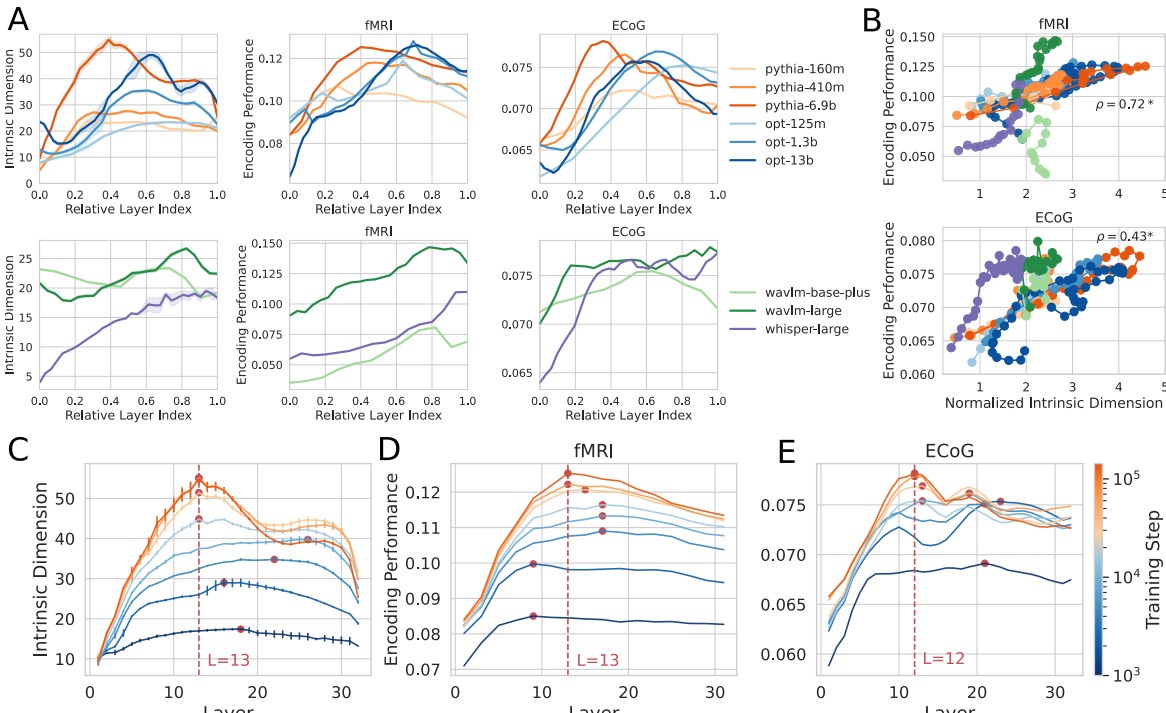

*Figure 2.* **Intrinsic dimension and encoding performance. (A) Layerwise $I_d$ accounts for layerwise encoding performance.** In LLMs and audio models, $I_d$ peaks in the deep layers, indicating a phase of high feature richness (left). This peak tracks the maximal layerwise encoding performance of both fMRI (middle) and ECoG (right). **(B) Global correlation between $I_d$ and encoding performance across model layers**. For fMRI (top) and ECoG (bottom), each model layer's encoding performance (y-axis) is shown against the $I_d$ (normalized by the log hidden dimension for comparison between models) (x-axis). The global correlation between the encoding performance and $I_d$ is $\rho = 0.76$ (fMRI) and $\rho = 0.43$ (ECoG), both significant at $\alpha = 0.05$. **(Bottom row) Encoding performance and $I_d$ peaks manifest concurrently over training**: **(C)** Pythia-6.9b's layerwise $I_d \pm 1$SD (over 5 random data splits) is shown over training, where an $I_d$ peak at layer 13 manifests over time. Curves correspond to training checkpoints 1K, 2K, 4K, 8K, 16K, 32K, 64K, and 143K (final checkpoint). **(D)** Training dynamics of layerwise encoding performance of the fMRI BOLD signal. A peak is reached at layer 13. **(E)** Training dynamics of layerwise ECoG encoding performance, where a peak is reached at layer 12. Red dots in each plot show maximal layers for the respective metric.

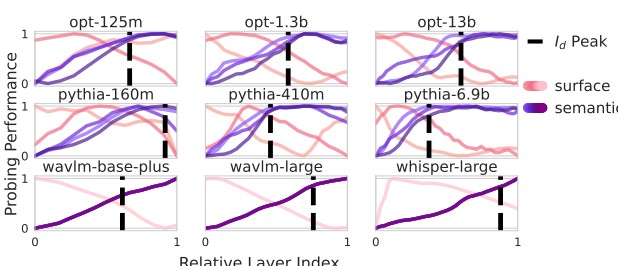

*Figure 3.* **Semantic decodability coalesces near the $I_d$ peak.** For LLMs (top rows) and speech models (bottom), higher-order feature decodability (purple) peaks in the deep layers near the $I_d$ peak (dashed line); superficial text or acoustic feature decodability (pink) drops over the layers. Task details in Appendix D.

across modalities, encoding performance and $I_d$ increase at similar rates over training, seen by similar positions of the checkpoint curves in the plots. fMRI encoding performance and $I_d$ are highly globally correlated with $\rho = 0.96$, and ECoG encoding performance and $I_d$ with $\rho = 0.64$ (both $p$-values <1e-3). Lastly, the location of the $I_d$ peak

(red dots, Figure 2C) changes over training, eventually settling at the same layers for peak encoding performance (red dots, Figure 2D and E). This rules out that the $I_d$ peak trivially reflects the Transformer architecture, e.g., layer index. Results suggest that the relationship between feature complexity, linguistic knowledge and brain predictivity emerges from exposure to language training data.

**Better predicted voxels and electrodes show a stronger relationship between $I_d$ and encoding performance** Figure 4 shows the voxels (fMRI Subject UTS03) and electrodes (all ECoG subjects) that were predicted above a threshold of $R = 0.2$ (top 33%) by some OPT-1.3b layer (see Figure J.2 for flatmaps of voxelwise encoding performance). These "well-predicted" voxels and electrodes largely fall in fronto-temporal regions shown to process language (Fedorenko et al., 2024); in contrast, voxels for which encoding performance was low ($R < 0.2$) largely fall outside of conventional language areas, for instance, in low-level visual areas.

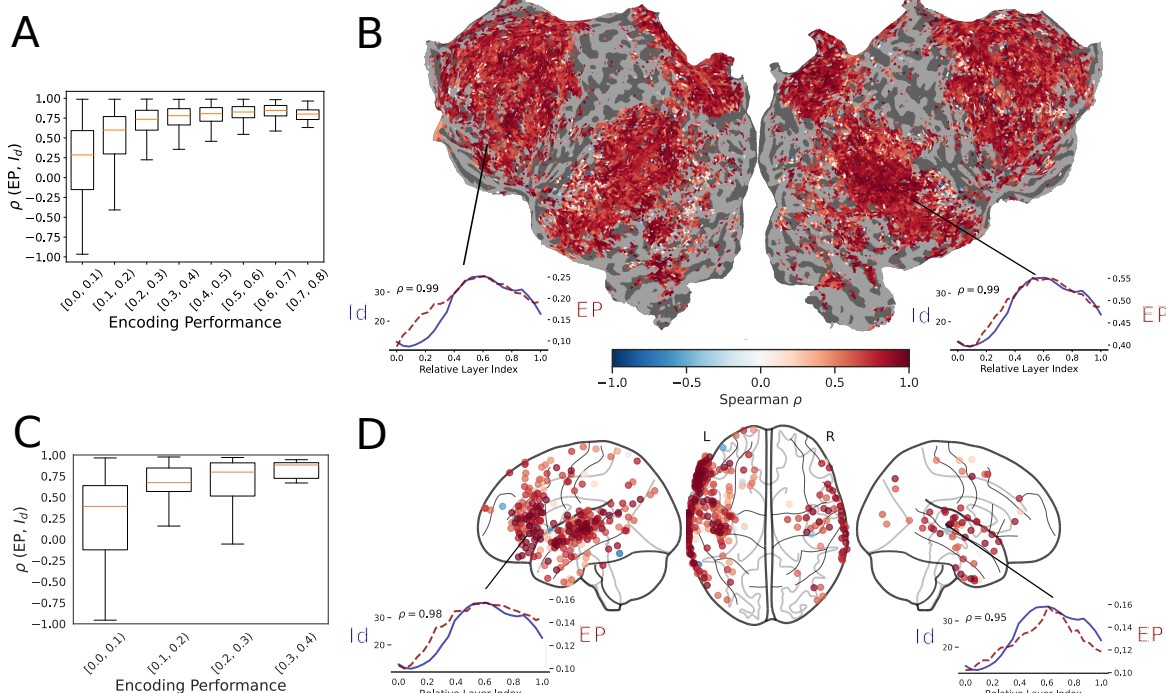

*Figure 4.* **How $\rho(I_d,$ encoding performance) distributes on the brain (OPT-1.3b). (A, C)** The Spearman $\rho(EP, I_d)$ between layerwise encoding performance (EP) and $I_d$ (y-axis) is plotted against encoding performance (x-axis) for **(A)** fMRI Subject UTS03 and **(C)** ECoG. Each point in the plot is a single voxel or electrode (95556 voxels, 1268 electrodes across subjects), where encoding performance is discretized into buckets for legibility. There is an increasing trend, $\rho = 0.58$ for fMRI and $\rho = 0.56$ for ECoG, $p <$1e-3), where voxels and electrodes that are better predicted by OPT-1.3b also have a stronger correlation between layerwise $I_d$ and encoding performance. **(B)** For fMRI Subject UTS03, we show, for voxels with sufficiently high encoding performance ($r \geq 0.2$), the Spearman $\rho$ between layerwise $I_d$ and encoding performance (red is higher). For these well-predicted voxels, layerwise $I_d$ correlates highly with encoding performance, $\rho = 0.73$. **(D)** For electrodes across subjects with high encoding performance ($r \geq 0.1$, roughly top 25% of electrodes), we plot the Spearman $\rho$ between layerwise $I_d$ and encoding performance (red is higher). For these electrodes, layerwise $I_d$ correlates positively to encoding performance, $\rho = 0.63$ on average. Correlations are highest in fronto-temporal language processing areas for both fMRI and ECoG. Several example voxels and electrodes' layerwise $I_d$ and encoding performance (EP) are shown (small plots).

Each voxel and electrode in Figure 4B,D is colored by the correlation between layerwise $I_d$ and encoding performance (dark red is better) for OPT-1.3b; we also show several highly-correlated example trajectories. For each model and subject, the encoding performance of well-predicted voxels grows with layerwise $I_d$ ($\rho = 0.73$ for OPT-1.3b fMRI Subject UTS03 and $\rho = 0.63$ for ECoG, see Table J.1 for other subjects and models). An exception is voxels near the primary auditory cortex (white and blue dots in Figure 4B), which process low-level auditory information; these voxels are predicted above $R = 0.2$ by LLMs and much better by speech model layers, but, they do not make use of abstract linguistic features in the high-$I_d$ middle layers. Still, in the vast majority of well-predicted voxels, $I_d$ correlates highly positively to encoding performance (much more red than blue in Figure 4B); further, the better-predicted a voxel or electrode is from language or speech models, the higher the correlation between layerwise encoding performance and $I_d$, see Figure 4A, where $\rho = 0.58$ for fMRI Subj. UTS03, ($p <$1e-3), and Figure 4C, where $\rho = 0.56$

for ECoG ($p <$1e-3); see Table J.2 for all subjects, modalities, and models. In sum, we see that in most voxels and electrodes, higher encoding performance tracks a stronger relationship to *abstraction* in the middle layers.

### 4.3. Increasing encoding performance increases intrinsic dimension and semantic content

Up to this point, our analyses linking $I_d$, feature richness, and encoding performance have been largely correlational. Now, as in Vattikonda et al. (2025), we causally manipulate the layerwise encoding performance by directly finetuning WavLM-base-plus—the model whose encoding performance was most driven by low-level input properties—on fMRI for each subject. See Appendix F for details.

Figure 5 shows that directly finetuning the best-performing WavLM layer (9) to better predict fMRI responses not only increases the encoding performance (left), but also increases the semantic content (middle) and the representational $I_d$ (right), where the original and braintuned versions are shown

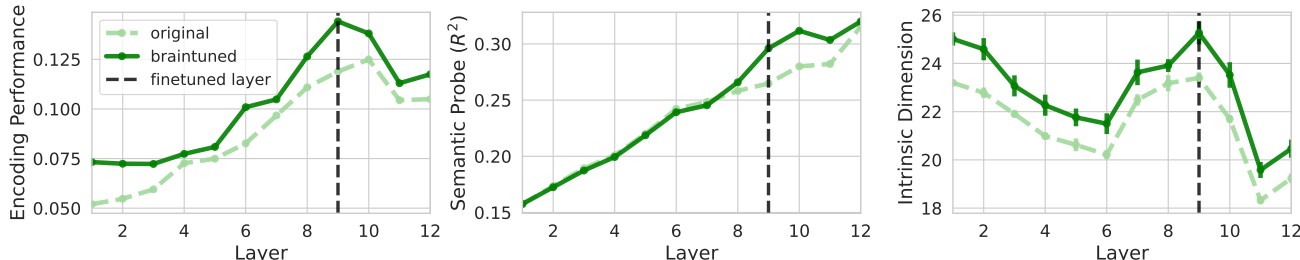

*Figure 5.* **Increasing encoding performance by finetuning increases semantic content and $I_d$.** Finetuning WavLM layer 9 on voxelwise responses to speech increases encoding performance (left), as well as semantic content (middle) and $I_d$ (right, light to dark green). Results are shown averaged over subjects; the error bars for $I_d$ (right) reflect $\pm$1SD aggregated across subjects and data splits.

in light and dark green, respectively. Results generalize for each subject, see Figure H.3. Our findings align with Vattikonda et al. (2025); Moussa et al. (2025), who found braintuning speech model representations to increase their semantic content; here, we add $I_d$ to the equation, having demonstrated a link in which higher overall similarity to the brain requires richer semantic content, which led to higher dimensional representations of language.

### 4.4. Dimensionality and encoding performance: mechanism or symptom?

We just saw that in language-audio models, encoding performance, $I_d$, and meaning abstraction are causally related. But is $I_d$ a *mechanism* or *symptom* in this relationship? Manipulating the $I_d$ of random feature spaces reveals that high $I_d$ is likely an epiphenomenon of learning rich features of language, some of which are useful for predicting the brain, rather than a reason for high predictivity in-and-of-itself.

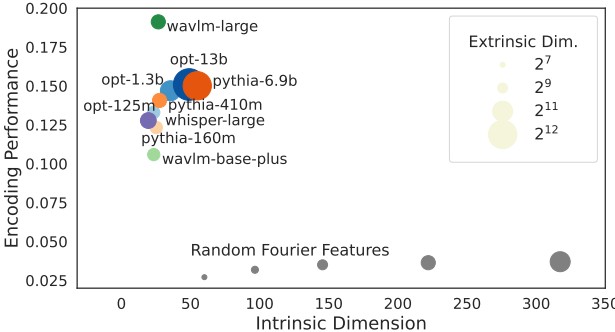

*Figure 6.* **Feature richness alone does not explain LLM and speech model encoding performance.** For the best LLM and speech layers (colors) as well as RFF spaces (gray) of increasing extrinsic dimension ($\circ$ size), we show the growth in $I_d$ (x-axis) with the corresponding growth in encoding performance averaged across fMRI subjects (y-axis).

**High-$I_d$ random features of linguistic inputs poorly predict the brain**   To test whether layer $I_d$ *per-se* influences model-brain predictivity, we constructed Random Fourier Feature spaces of increasing $I_d$, measured on The Pile for

comparison to language-audio models. Using these feature maps, we embedded the same stimuli to the human subjects, learning an encoding model from RFF space to brain responses. If $I_d$ alone causally drives encoding performance, then the RFF map should predict brain responses as well as an LLM or speech model layer with the same $I_d$.

Figure 6 shows how fMRI encoding performance (y-axis) grows with $I_d$ (x-axis) for random Fourier features and the best-performing language-audio model layers across any subject. There is a perfect correlation, $\rho = 1$, between RFF-$I_d$ and encoding performance, but the effect is small: Figure 6 shows significant diminishing returns to increasing the RFF-$I_d$ beyond 150, where final encoding performances plateau at $R \approx 0.04$. Meanwhile, an LLM or speech model layer with lower $I_d$ achieves encoding performances of $R > 0.1$ (figure top left). Findings hold for ECoG, see Figure I.1. Results show that $I_d$ does not *cause* model-brain predictivity. Instead, it is the other way around: the high $I_d$ of the best encoding layers arises from learning rich abstractions of language.

## 5. Discussion

Recent studies on language encoding models have observed that the intermediate layers of speech models and LLMs, rather than the output layers, are most linearly similar to measured brain activity (Antonello & Huth, 2022; Hong et al., 2024; Caucheteux et al., 2023). Despite this frequently observed trend, little research has been dedicated to explaining it. LLM and speech model layers are invariant to many variables: each layer has the same architecture and was trained using the same data and downstream objective. Then, layer-wise differences can only arise either due to the abstractive nature of the transition from earlier to later layers, or due to the "time pressure" exerted by the loss term on later layers. These competing pressures, to first build up a comprehensive representation of the input text, and to then resolve this representation towards a distribution over predicted next token outputs, have opposite effects, as we demonstrate here. Meaning abstraction leads

to an increase in encoding performance and dimensionality, whereas prediction narrows dimensionality to the detriment of encoding.

What conclusions should we draw from this? First, that it is unlikely that the autoregressive nature of language models directly drives brain-model similarity (Schrimpf et al., 2021; Goldstein et al., 2022; Antonello & Huth, 2022). As models become more potent at prediction, their most predictive and most descriptive layers drift apart. Secondly, our results seem to support the claim from other works (Cheng et al., 2025; Skean et al., 2025; Lad et al., 2025) that LLMs have an intrinsic two-phase process, specifically, a first phase that supports composition and abstraction, followed by a second "output" phase that focuses on prediction.

Putting these conclusions together with the existing literature, we can update the story of the reasons for model-brain alignment. Contemporary LLMs trained on next-token prediction better predict neural responses than static embedding models like word2vec (Caucheteux & King, 2022; Schrimpf et al., 2021; Goldstein et al., 2022). In these performant models, the high-$I_d$ intermediate layers, not the final predictive layers, best predict the brain; these high-$I_d$ layers correspond to a phase of higher-order linguistic feature-building about the inputs. In language models, these feature-rich layers do eventually serve next-token prediction, once they are processed by the final output layers after this abstraction peak.

Finally, we observe that representations' $I_d$ explains their encoding performance *in the context of* deep models trained to embed the structure of text or speech. As our random Fourier feature analysis demonstrates, high $I_d$ in-and-of-itself is necessary but not sufficient for high neural predictivity. But the observation that braintuning an autoregressive speech model directly increases its $I_d$ implies that, although prediction objectives can learn some of the structure that contributes to neural alignment, they still leave out readily learnable nonlinear features that are useful for predicting brain activity. This heavily implies that the nonlinearities learned by LLMs through autoregressive loss provide only an incomplete picture of the nature of learning and intelligence. Therefore, developing better methods for extracting task-relevant structure from naturalistic data may be key to advancing encoding models beyond their current limits and, in turn, improving what these models can tell us about the underlying mechanisms of the brain.

## Limitations

We note that our work does not fully provide a detailed interpretable account of which specific features $I_d$ tends to encode in language and speech models, such as acoustic, semantic, or syntactic features. Intrinsic dimension is

a coarse-grained, dataset-level measure that indicates *how many* degrees of freedom underlie a dataset, but does not tell us what those degrees of freedom are. Attributing $I_d$ to explicit and interpretable features is an important problem that remains open in the $I_d$ estimation literature. Layer-wise probing alleviates this to some extent, where we found higher $I_d$ to mark a phase of higher-order linguistic processing. Finding an explicit description of the degrees of freedom underlying model-brain similarity is an important direction for future work, though Kauf et al. (2024) have already suggested encoding of lexical semantics to play a privileged role. Moreover, while our brain-finetuning results provide a critical element of causal analysis to our work, we do not purport to present the full causal story that relates intrinsic dimensionality to encoding performance. As we have noted, such analyses are rife with co-dependencies that are difficult to disentangle. Further work is necessary to fully resolve this picture.

## Impact Statement

This paper addresses the reasons for predictivity of neural responses from deep neural network representations. The insights can be used to better understand how relatively opaque systems like deep neural nets and the brain process language.

### Acknowledgments

This project has received funding from the European Research Council (ERC) under the European Union's Horizon 2020 research and innovation programme (grant agreement No. 101019291) and was additionally supported by NIDCD grant R01DC014279. Computing resources were provided by the Zuckerman Institute at Columbia University and the Texas Advanced Computing Center. This paper reflects the authors' view only, and the funding agency is not responsible for any use that may be made of the information it contains.

We would like to thank the Center for Brains, Minds, and Machines for hosting the collaboration, and Marco Baroni, Alex Huth, Nima Mesgarani, Alessandro Laio, and members of the UPF linguistics department for feedback.

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

# A. Data, Code, and Computational Resources

## A.1. Code

https://github.com/chengemily1/brain-id-abstract.

**DadaPy** https://github.com/sissa-data-science/DADApy; license: apache-2.0

**Patchscopes** https://pair-code.github.io/interpretability/patchscopes/; license: apache-2.0

**Surprisal** https://github.com/aalok-sathe/surprisal; license: MIT

**Encoding models** https://github.com/HuthLab/encoding-model-scaling-laws; license: Unknown

## A.2. Models

**WavLM** e.g., https://huggingface.co/microsoft/wavlm-large; license: cc

**Whisper** e.g., https://huggingface.co/openai/whisper-large; license: apache-2.0

**OPT** e.g., https://huggingface.co/facebook/opt-13b; license: OPT-175B

**Pythia** e.g., https://huggingface.co/EleutherAI/pythia-6.9b-deduped; license: apache-2.0

## A.3. Data

Data were derived from the following open-source datasets.

**The Pile sample** https://huggingface.co/datasets/NeelNanda/pile-10k; license: bigscience-bloom-rail-1.0

**LibriSpeech ASR** https://huggingface.co/datasets/openslr/librispeech_asr; license: cc by 4.0

**Probing tasks (Conneau et al., 2018)** https://github.com/facebookresearch/SentEval/tree/main/data/probing; license: bsd

**fMRI (LeBel et al., 2023)** https://openneuro.org/datasets/ds003020/versions/2.0.0; license: cc0

**ECoG (Zada et al., 2025)** https://openneuro.org/datasets/ds005574; license: cc0

## A.4. Compute

Ridge regression was performed using compute nodes with 128 cores (2 AMD EPYC 7763 64-core processors) and 256GB of RAM. In total, roughly 1,000 node-hours of compute was expended for these models. Feature extraction for language models and braintuning for speech models were performed on specialized GPU nodes similar to the AMD compute nodes but with 3 NVIDIA A100 40GB cards. Feature extraction required roughly 300 node-hours of compute on these GPU nodes, and braintuning required roughly 50 node-hours.

Dimensionality and surprisal computation were run on a cluster with 12 nodes with 5 NVIDIA A30 GPUs and 48 CPUs each. Extracting and computing dimensionality on LM representations took a few wall-clock hours per model. Training TunedLens took around 15 minutes per layer, so overall 30 wall-clock hours. Training the layerwise linguistic feature probes took around 1 hour per model per task, so a total of around 60 hours. We parallelized all computation, and estimate the overall parallelized runtime, including preliminary experiments and failed runs to be around 40 days.

**AI Disclosure** ChatGPT was used for minor coding assistance. AI was not used in any capacity in writing.

## B. fMRI Stimuli

We list all the names of the stimuli used for training (95 stories) and testing (3 stories) the fMRI encoding models. The complete list can be found in LeBel et al. (2023).

**Train stories (N=95)** itsabox, odetostepfather, inamoment, afearstrippedbare, findingmyownrescuer, hangtime, ifthishair-couldtalk, goingthelibertyway, golfclubbing, thetriangleshirtwaistconnection, igrewupinthewestborobaptistchurch, tetris, becomingindian, thetiniestbouquet, swimmingwithastronauts, lifereimagined, forgettingfear, stumblinginthedark, backsideofthestorm, food, theclosetthatateeverything, escapingfromadirediagnosis, notontheusualtour, exorcism, adventuresinsayingyes, thefreedomridersandme, cocoonoflove, waitingtogo, thepostmanalwayscalls, googlingstrangersand-kentuckybluegrass, mayorofthefreaks, learninghumanityfromdogs, shoppinginchina, souls, cautioneating, comingofa-geondeathrow, breakingupintheageofgoogle, gpsformylostidentity, marryamanwholoveshismother, eyespy, treasureisland, thesurprisingthingilearnedsailingsoloaroundtheworld, theadvancedbeginner, goldiethegoldfish, life, thumbsup, seedpotatoesofleningrad, theshower, adollshouse, sloth, howtodraw, quietfire, metsmagic, penpal, thecurse, canadageese-andddp, thatthingonmyarm, buck, thesecrettomarriage, wildwomenanddancingqueens, againstthewind, indianapolis, al-ternateithicatom, bluehope, kiksuya, afatherscover, haveyoumethimyet, firetestforlove, catfishingstrangerstofindmyself, christmas1940, tildeath, lifeanddeathontheoregontrail, vixenandtheussr, undertheinfluence, beneaththemushroomcloud, jugglingandjesus, superheroesjustforeachother, sweetaspie, naked, singlewomanseekingmanwich, avatar, whenmothers-bullyback, myfathershands, reachingoutbetweenthebars, theinterview, stagefright, legacy, listo, gangstersandcookies, birthofanation, mybackseatviewofagreatromance, lawsthatchokecreativity, threemonths, whyimustspeakoutaboutclimat-echange, leavingbaghdad

**Test stories (N=3)** wheretheressmoke, fromboyhoodtofatherhood, onapproachtopluto

## C. ID Estimation

### C.1. Nonlinear intrinsic dimension

We use a state-of-the-art $I_d$ estimator, the Generalized Ratios Intrinsic Dimension Estimator (GRIDE) (Denti et al., 2022). GRIDE highly correlates to other estimators (Binnie et al., 2025) such as TwoNN (Facco et al., 2017) and the Maximum Likelihood Estimator (MLE) (Levina & Bickel, 2004) while relaxing their assumptions on local uniformity.

In brief, manifold dimension estimators like GRIDE, TwoNN, and MLE rely on the fact that manifolds are locally Euclidean (Campadelli et al., 2015). This permits estimating the intrinsic dimension of the manifold based on local neighborhoods. Estimators such as TwoNN (Facco et al., 2017), local Principal Component Analysis (Fukunaga & Olsen, 1971), and the Maximum Likelihood Estimator (Levina & Bickel, 2004) are sensitive to *scale*, i.e. how "local" those neighborhoods are (Facco et al., 2017; Denti et al., 2022). Imposing certain locality assumptions such as uniform density up to the second nearest neighbor (Facco et al., 2017), it is then possible to derive a theoretical distribution over local distances, volumes, or angles. Then, the $I_d$ can be recovered via maximum likelihood estimation.

The Generalized Ratios Intrinsic Dimension Estimator (GRIDE) is a methodological successor to the TwoNN estimator (Facco et al., 2017) that relaxes TwoNN's assumption of uniformity up to the second nearest neighbor. GRIDE instead produces unbiased estimates up to the $2k^{th}$ nearest neighbor. This permits an analysis of the $I_d$'s dependence on the scale $k$.

In particular, GRIDE operates on ratios $\mu_{i,2k,k} := r_{i,2k}/r_{i,k}$, where $r_{i,j}$ is the Euclidean distance between point $i$ and its $j^{th}$ neighbor. Assuming local uniform density up to the $2k^{th}$ neighbor, the ratios $\mu_{i,2k,k}$ follow a generalized Pareto distribution

$$f_{\mu_i,2k,k}(\mu) = \frac{I_d(\mu^{I_d} - 1)^{k-1}}{B(k,k)\mu^{I_d(2k-1)+1}},$$ (C.1)

where $B(\cdot,\cdot)$ is the beta function. The $I_d$ is then recovered by maximizing this likelihood over points $i$ for several candidate scales $k$. Finally, in order to choose the proper $I_d$, a scale analysis over $k$, which controls the neighborhood size, is necessary: if $k$ is too small, the $I_d$ likely describes local noise, and if $k$ is too large, the curvature of the manifold will produce a faulty estimate. Instead, it is recommended to choose a $k$ for which the $I_d$ is stable (Denti et al., 2022).

For $I_d$ estimation using GRIDE, we reproduce the setup in Cheng et al. (2025). For each model, checkpoint, and layer, we perform a scale analysis. Figure C.1 shows an example, where the GRIDE scale $k$ varies from $2^0$ to $2^{12}$. As recommended in Denti et al. (2022), we choose a scale $k$ corresponding in a range where the intrinsic dimension is stable, or plateaus, by

*Table C.1.* Selected GRIDE scales $k$ after performing a scale analysis for intrinsic dimension estimation, for all models and checkpoints tested.

| Model | GRIDE $k$ |
|---|---|
| OPT-125m | 64 |
| OPT-1.3b | 32 |
| OPT-13b | 32 |
| Pythia-160m | 128 |
| Pythia-410m | 128 |
| Pythia-6.9b | 16 |
| Pythia-6.9b ($t =$64000) | 16 |
| Pythia-6.9b ($t =$32000) | 32 |
| Pythia-6.9b ($t =$16000) | 32 |
| Pythia-6.9b ($t =$8000) | 32 |
| Pythia-6.9b ($t =$4000) | 64 |
| Pythia-6.9b ($t =$2000) | 16 |
| Pythia-6.9b ($t =$1000) | 16 |
| WavLM-base-plus | 2 (first 5 layers), 1 after |
| WavLM-base-plus (finetuned on fMRI subj. UTS02) | 2 (first 5 layers), 1 after |
| WavLM-base-plus (finetuned on fMRI subj. UTS03) | 2 (first 5 layers), 1 after |
| WavLM-large | 2 (first 8 layers), 1 after |
| Whisper-large | 16 |

*Table C.2.* **Robustness of $I_d$ peak layer to $k$.** The location of the central $I_d$ peak is quite robust to the choice of $k$. Each entry in the table is the layer index of the ID peak, shown for the original choice of $k$ as well as for $k \div / \times 2$. Halving and doubling $k$, which drastically change the neighborhood size, do not considerably affect the location of the peak. The largest changes (4 layers) are in Pythia-410m and Whisper-large, however, over these layers the $I_d$ curve is relatively flat and the shape of the $I_d$ curve is not affected, see Figure C.2.

| | OPT | | | Pythia | | | WavLM | | Whisper |
|---|---|---|---|---|---|---|---|---|---|
| | 125m | 1.3b | 13b | 160m | 410m | 6.9b | base-plus | large | large |
| $k$ | 9 | 15 | 25 | 3 | 12 | 13 | 9 | 20 | 30 |
| $k \div 2$ | 11 | 15 | 26 | 4 | 11 | 13 | 8 | 19 | 26 |
| $k \times 2$ | 9 | 15 | 23 | 5 | 16 | 13 | 6 | 19 | 30 |
| # layers | 12 | 24 | 40 | 12 | 24 | 32 | 12 | 24 | 36 |

visual inspection. For simplicity, we choose one scale $k$ per model, for instance, in the particular example in Figure C.1, we choose $k = 2^4$, where the derivative of the curve is closest to 0 for as many layers as possible. WavLM had a notable shift in scaling behavior between early and late layers; for WavLM (base and large), we choose one scale $2^1$ for the first third of model layers, and $2^0$ for the rest. Scales chosen for all models are in Table C.1.

### C.1.1. ROBUSTNESS OF RESULTS TO CHOICE OF GRIDE SCALE $k$

Changing $k \div / \times 2$ does not considerably change the $I_d$ curve, see Figure C.2, nor the peak layer index, see Table C.2. This is by design, as $k$ was chosen such that changes in $k$ would result in minimal changes in $I_d$.

## C.2. Linear dimensionality

We computed the linear dimension two ways: PCA with a variance cutoff of 0.99, and the Participation Ratio. After centering representations, we compute the eigenspectrum $\lambda_1 \geq \lambda_2 \geq \cdots \lambda_D$ of its representations' covariance matrix. The dimensionality given by PCA with a threshold of 0.99 is given by the minimal number of principal components that explain at least 99% of the variance. The Participation Ratio (PR), a non-integer measure of the effective dimension, is given by

$$d_{PR}(X) = (\sum_j \lambda_j)^2 / (\sum_j \lambda_j^2). \tag{C.2}$$

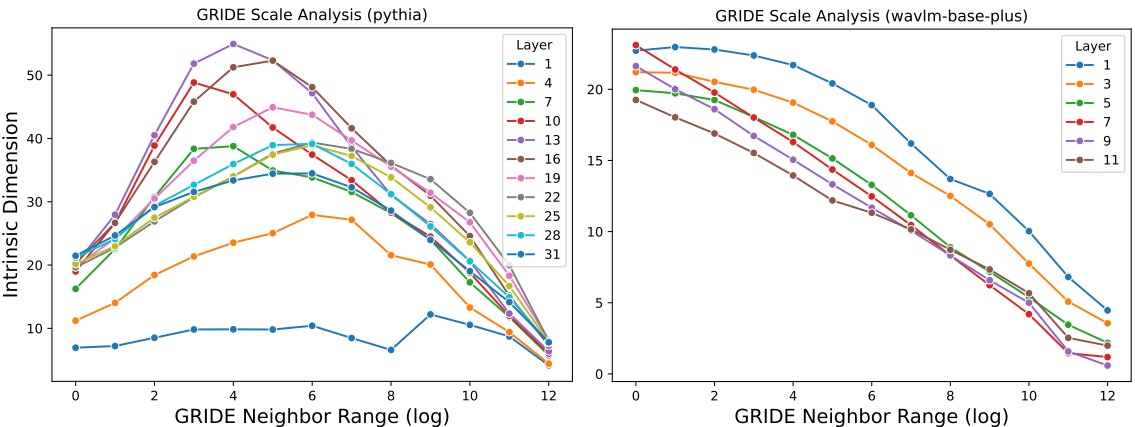

*Figure C.1.* Example GRIDE scale analysis for Pythia-6.9b and WavLM-base-plus. The estimated intrinsic dimension (y axis) varies according to the chosen scale $k$ (x axis). It is recommended to choose a scale where the local change is minimal. In the case of Pythia (left), we chose $k = 2^4$, and in the case of WavLM-base-plus (right), we chose $k = 2^1$ for layers up to and including layer 5, and $k = 2^0$ after.

*Figure C.2.* $I_d$ **curve for different GRIDE scales** $k$. For each model, we show the original $I_d$ curve (solid lines) found by the scale analysis, along with $k \div / \times 2$ (dashed lines). All curves are the average of 5 random data splits. Qualitatively, the shape of the curve does not considerably change despite the drastic change in neighborhood size. All models display a middle-late layer rise in the $I_d$, which occurs no matter the choice of $k$.

*Table C.3.* **Robustness of encoding performance to choice of** $k$**.** For fMRI and ECoG (averaged over participants), we show how the $\rho(I_d, \text{EP})$ correlation changes when changing $k$ ($\div$, $\times 2$). Overall, qualitative trends remain the same—$I_d$ correlates highly with encoding performance for most models except for WavLM-base-plus and moderately for Pythia-160m and speech models, as discussed in the main paper. For WavLM-base-plus, for which correlations were not significant, there were larger changes in correlation with respect to $k$. However, for the vast majority of models already displaying high correlation, the change was negligible.

| Modality | Scale | OPT 125m | 1.3b | 13b | Pythia 160m | 410m | 6.9b | WavLM base-plus | large | Whisper large |
|---|---|---|---|---|---|---|---|---|---|---|
| | $k$ | 0.90* | 0.82* | 0.85* | 0.33 | 0.91* | 0.92* | 0.02 | 0.93* | 0.84* |
| fMRI | $k \div 2$ | 0.69* | 0.90* | 0.88* | 0.32 | 0.92* | 0.84* | 0.03 | 0.90* | 0.88* |
| | $k \times 2$ | 0.85* | 0.83* | 0.83* | 0.29 | 0.86* | 0.95* | -0.38 | 0.95* | 0.79* |
| | $k$ | 0.97* | 0.97* | 0.82* | 0.83* | 0.88* | 0.88* | 0.21 | 0.42* | 0.64* |
| ECoG | $k \div 2$ | 0.97* | 0.93* | 0.82* | 0.83* | 0.96* | 0.88* | 0.46 | 0.41* | 0.60* |
| | $k \times 2$ | 0.97* | 0.92* | 0.92* | 0.86* | 0.85* | 0.87* | 0.15 | 0.43* | 0.62* |

The PR is designed to smoothly interpolate between 1 and $D$: one can verify that when $\lambda_{i \neq 1} = 0$, then $d_{PR}(X) = 1$, and when data are isotropic, that is, $\lambda_i = \lambda_j \ \forall i \neq j$, then $d_{PR}(X) = D$ (Gao et al., 2017).

### C.3. Results using linear dimensionality

Table C.4 shows the correlation between linear dimension computed with PCA and PR to the fMRI encoding performance for subject UTS03, for several models. In general, the correlations were high, and comparable to the correlations between $I_d$ and layerwise encoding performance. However, the dimensionalities themselves poorly correlated to semantic probing tasks, and Pythia-6.9b in particular showed a low overall correlation between $I_d$ and encoding performance. The visual comparison between $I_d$ and PR is shown in Figure C.3 for the average fMRI encoding performance across subjects and voxels (top), as well as several example voxels (bottom).

Tuckute et al. (2023) found a weak positive correlation between the linear effective dimension (PR) of the embedded training set stimuli and their encoding performance in audio models. This, however, is different from our setting as we use the dimensionality on *generic stimuli* as a way to localize abstract processing stages in deep language-audio models.

*Table C.4.* The average voxelwise product-moment correlations between representational dimensionality and encoding performance are shown for PCA-$d$ (variance threshold of 0.99), and PR-$d$. Across models, the correlation is generally high no matter the dimensionality measure, though PR in particular did not behave consistently across models (see Pythia-6.9b). All values, except those marked with (*), are significant to $p < 10^{-3}$, as computed by a permutation test.

| | OPT-125m | OPT-1.3b | OPT-13b | Pythia-6.9b |
|---|---|---|---|---|
| PCA $d$ | 0.91 | 0.93 | 0.96 | 0.86 |
| PR $d$ | 0.94 | 0.82 | 0.85 | -0.05* |

## D. Layerwise Probing Experiments

### D.1. LLM probing experiments using SentEval, Conneau et al. (2018)

Like Cheng et al. (2025); Baroni et al. (2026), we use the SentEval dataset (Conneau et al., 2018) for layerwise probing experiments. SentEval consists of a number of classification tasks given sentence representations. Tasks in SentEval span from surface-level (e.g., selecting whether a word appears in the input) to higher-order (e.g., choosing whether the two coordinate clauses in a sentence are inverted). The list of tasks we use and their description are given in the below table with their categorizations into whether they require surface-level or potentially higher-order interpretations (labeled "semantic" in Figure 3) of the inputs.

We use the last token residual stream representation for decoding, consistent with the rest of the experiments. Each task is trained using 100k examples, a validation set of 10k examples, and tested on a test set of 10k datapoints. Linear probes are trained using Adam (default Pytorch parameters), with a learning rate of $5e$-3, over 15 epochs. The best linear probe by validation accuracy over the 15 epochs is selected as the final model, which is then evaluated on the test set.

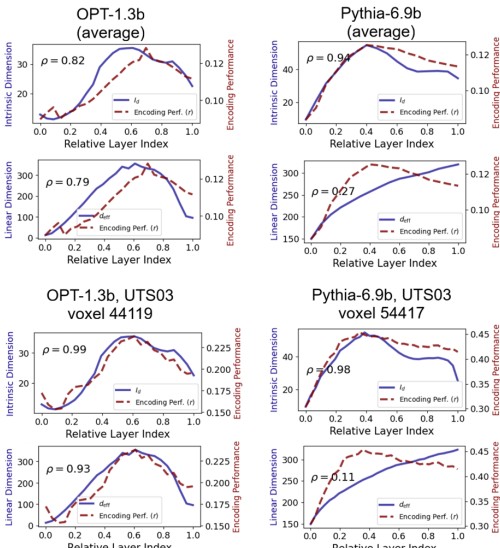

*Figure C.3.* **Examples of $I_d$ vs. linear effective dimension (PR) in explaining fMRI encoding performance.** Using the Participation Ratio as a dimensionality measure did not outperform $I_d$ in predicting encoding performance, shown here for fMRI, averaged over all voxels and subjects at the top and for single voxel examples at the bottom. In the "well-behaved" case of OPT-1.3b (left column), $I_d$ correlates to the PR; however, for Pythia-6.9b, PR generally *increases* over layers as opposed to exhibiting the bell shape characteristic of $I_d$.

| Task | Category | Description |
|---|---|---|
| Word Content | Surface | Selecting whether a word appears in the sentence via 1000-way classification, where the 1000 indices correspond each to a word. |
| Sentence Length | Surface | A 3-way classification into short, medium, or long sentences. |
| Bigram Shift | Higher-order | Binary classification of whether two consecutive tokens in the sentence were inverted. |
| Odd Man Out | Higher-order | Binary classification of whether a noun in the original sentence has been replaced by a random other noun. |
| Coordination Inversion | Higher-order | Binary classification of whether two coordinate clauses were inverted. |

### D.2. Speech audio model probing using the features in Vattikonda et al. (2025)

We replicate the linguistic probing setup of Vattikonda et al. (2025), probing layerwise along each speech model for acoustic and semantic representations (Pasad et al., 2021; Vaidya et al., 2022). For acoustic content, we filter the audio waveform with a Mel filterbank, and predicted it with ridge regression from the intermediate layer representations. To probe semantic content at each layer, we predicted, also using ridge regression, the 300-dimensional GloVe embeddings (Pennington et al., 2014) of the current word being spoken. The current word was extracted using the timestamped transcripts from the original LeBel et al. (2023) dataset. Probes were evaluated using variance explained ($R^2$).

## E. Layerwise Surprisal Estimation

**LLMs** For LLMs, we used the TunedLens (Belrose et al., 2023) implementation by Ghandeharioun et al. (2024). TunedLens ascertains the amount of information (linearly) encoded in hidden layer $t$ about the next token. To do so, an affine mapping is learned from the last-token hidden representation at layer $t$ to the last hidden representation. In the provided code (Ghandeharioun et al., 2024), TunedLens is implemented using a direct solver `numpy.linalg.lstsq` on the training set. Finally, we compute the next-token surprisals on a validation set drawn from the same distribution. The training set is $N = 8000$ randomly sampled sequences from The Pile dataset (Gao et al., 2020), and the test set $N = 2000$ sequences

from The Pile.

**Speech audio models** For speech audio models, we had to construct our own train and test sets using Librispeech (Panayotov et al., 2015). We used the `train.clean.360` subset from HuggingFace and the Whisper-small automatic speech recognizer to transcribe each audio chunk ($N = 140k$, ranging from 1-20 seconds) to text with word timestamps. For each audio chunk, we randomly selected a word index between 1 and 5 from the end to truncate the audio for next-token prediction. We then tokenized the next word using the Whisper tokenizer and took the first token (using the Whisper tokenizer for WavLM, as WavLM does not have a native tokenizer). Finally, using the $N = 140k$ (audio chunk, next token) pairs, we trained affine maps to the Whisper vocabulary space. We evaluate the trained maps on the `test.clean` subset from HuggingFace, $N \approx 2000$. The maps were trained via Adam (default Pytorch hyperparameters) with a learning rate of 1e-4. We reported the best test loss (the average surprisal over tokens) over 10 epochs of training.

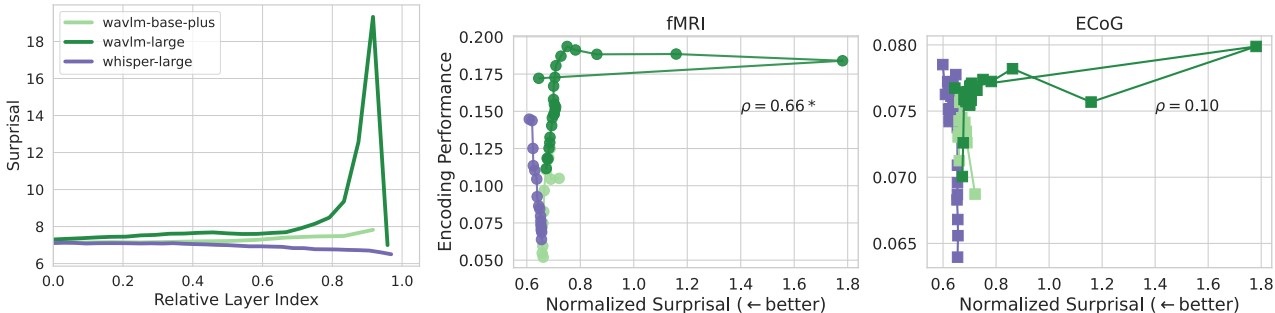

*Figure E.1.* **Surprisal and Encoding Performance for Audio Models.** Companion to Figure 1. The next-token prediction error on LibriSpeech for WavLM and Whisper models. There is a general decreasing trend where deeper layers predict the next token slightly better, but an exception is WavLM-large (dark green). **(Middle, Right)** The encoding performance averaged across fMRI (middle) and ECoG subjects (right) is plotted against normalized surprisal (surprisal divided by $\log$ Whisper's vocab size). There is a positive correlation in both cases between encoding performance and normalized surprisal, which provides evidence against the hypothesis that representations that are more predictive of the next token are also more predictive of the brain.

## F. Finetuning WavLM on brain responses

We follow the procedure of Vattikonda et al. (2025) to finetune WavLM models on an fMRI encoding task through the 9th layer of the models. Starting from the `wavlm-base-plus` checkpoint, we finetuned separate models for subjects UTS01, UTS02, and UTS03, and we extend the finetuning context from the original implementation of $2\,\mathrm{s}$ of audio to $4\,\mathrm{s}$.

## G. Additional Results on $I_d$ and Encoding Performance

*Table G.1.* **Max layer index** for $I_d$, fMRI and ECoG encoding performance averaged across subjects. The layers around the $I_d$ peak are the best for encoding, as seen by the small distance between the first column and next two columns compared to the number of layers (right column).

| Model | $I_d$ peak | fMRI | ECoG | N layers |
|---|---|---|---|---|
| opt-125m | 9 | 8 | 9 | 12 |
| opt-1.3b | 15 | 17 | 15 | 24 |
| opt-13b | 25 | 29 | 24 | 40 |
| pythia-160m | 4 | 4 | 6 | 12 |
| pythia-410m | 12 | 12 | 11 | 24 |
| pythia-6.9b | 13 | 14 | 11 | 32 |
| wavlm-base-plus | 9 | 10 | 7 | 12 |
| wavlm-large | 20 | 19 | 22 | 24 |
| whisper-large (encoder only) | 30 | 32 | 31 | 32 |

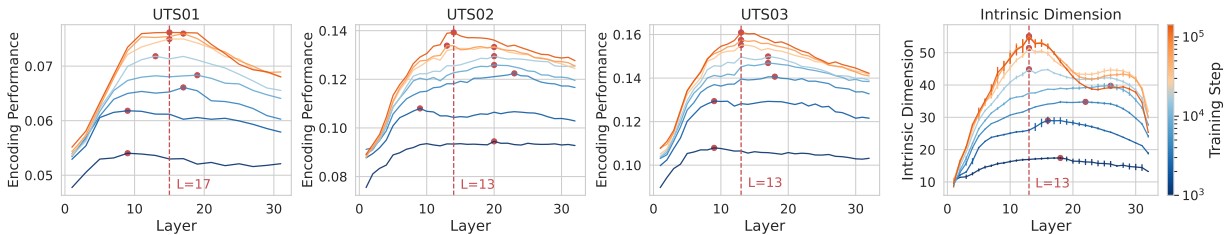

*Figure H.2.* **Training dynamics of layerwise encoding performance and $I_d$, Pythia-6.9b (each subject).**

# H. Per-subject results for fMRI

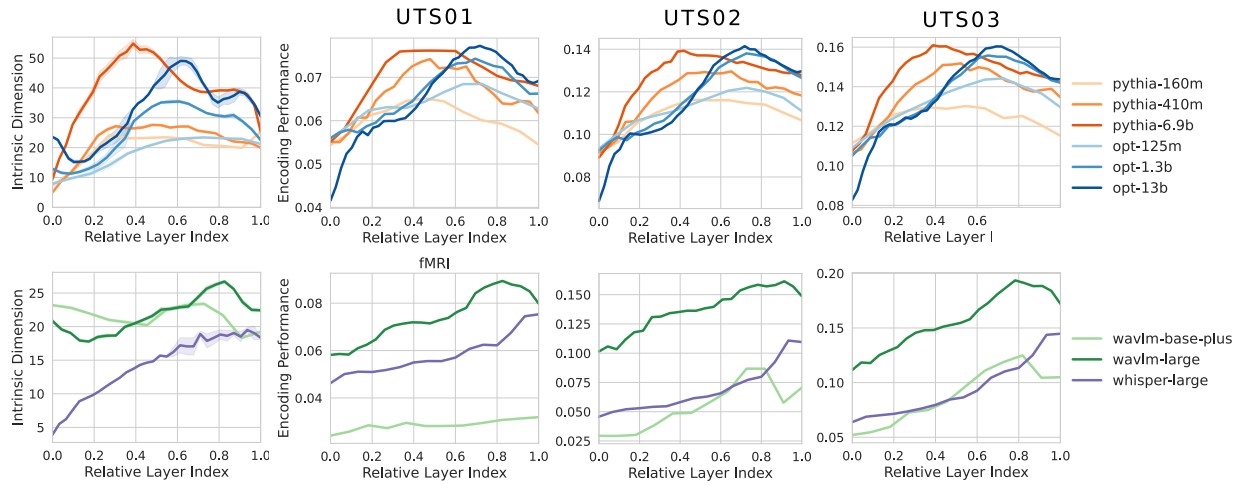

*Figure H.1.* **fMRI per-subject breakdown of how $I_d$ relates to encoding performance**

*Table H.1.* **Correlations between surprisal, $I_d$, and encoding performance for fMRI subjects UTS01-3** The average voxelwise product-moment correlations between representational dimensionality and encoding performance are shown for layerwise surprisal computed with the TunedLens (top row) and for $I_d$ (bottom row). For almost all models, the Spearman correlation between surprisal and encoding performance is generally not significant or negative, while the correlation between $I_d$ and encoding performance is significantly positive. Values significant with a $p$-value cutoff of 0.05, as determined by a permutation test, are marked with *.

| **Subj.** | | | OPT | | | Pythia | | | WavLM | | Whisper | LLMs | Speech | All |
| --- | --- | --- | --- | --- | --- | --- | --- | --- | --- | --- | --- | --- | --- |
| | | 125m | 1.3b | 13b | 140m | 410m | 6.9b | base | large | large | | | |
| UTS01 | Surprisal ↓ | -0.63* | -0.75* | -0.82* | 0.07 | -0.24 | -0.22 | 0.74* | 0.73* | **-0.99***| -0.57 | **-0.40***| **-0.77***|
| | $I_d$ ↑ | **0.92***| **0.90***| **0.85***| **0.51** | **0.93***| **0.95***| -0.52 | **0.86***| 0.97* | **0.89***| -0.43 | 0.68* |
| UTS02 | Surprisal ↓ | $-0.76^*$ | **-0.88***| **-0.85***| -0.47 | $-0.48^*$ | $-0.33^*$ | 0.84* | 0.83* | **-0.99** | -0.69* | 0.66 | -0.32* |
| | $I_d$ ↑ | **0.98***| 0.81* | **0.83***| **0.48** | **0.86***| **0.85***| -0.04 | **0.88***| **0.99** | **0.88***| **0.45***| **0.73***|
| UTS03 | Surprisal ↓ | -0.66* | -0.80* | -0.80* | -0.04 | -0.39 | -0.17 | 0.82* | 0.76* | **-0.99***| -0.57* | 0.66 | -0.51* |
| | $I_d$ ↑ | **0.92***| **0.86***| **0.88***| **0.49** | **0.90***| **0.95***| -0.09 | **0.90***| 0.98* | **0.90***| **0.51***| **0.76***|

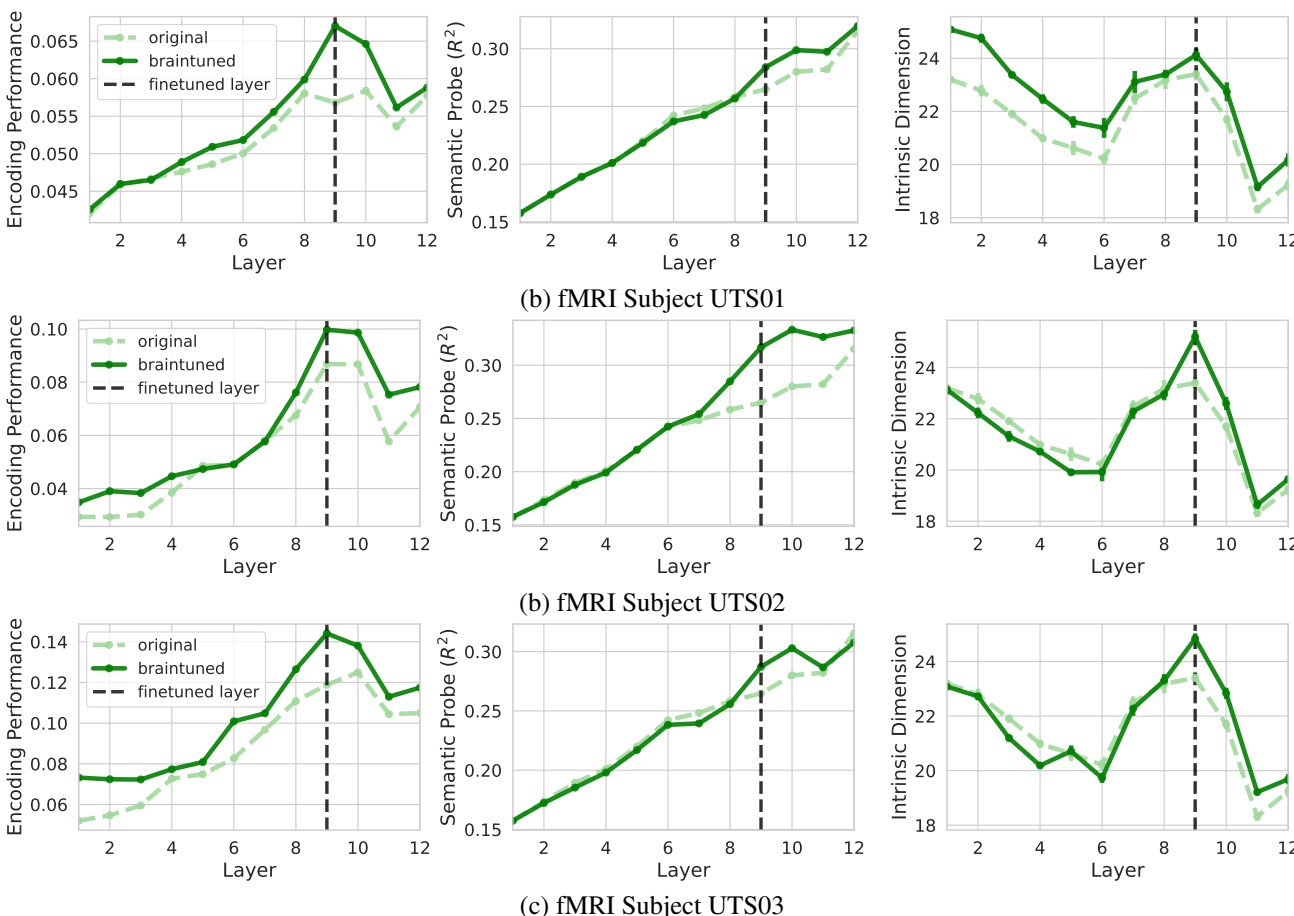

*Figure H.3.* **Increasing encoding performance by finetuning increases $I_d$ for each subject.** Companion to Figure 5. After finetuning WavLM-base-plus on voxelwise responses to speech on layer 9, the semantic content of representations (middle) and layerwise intrinsic dimension (right) increase.

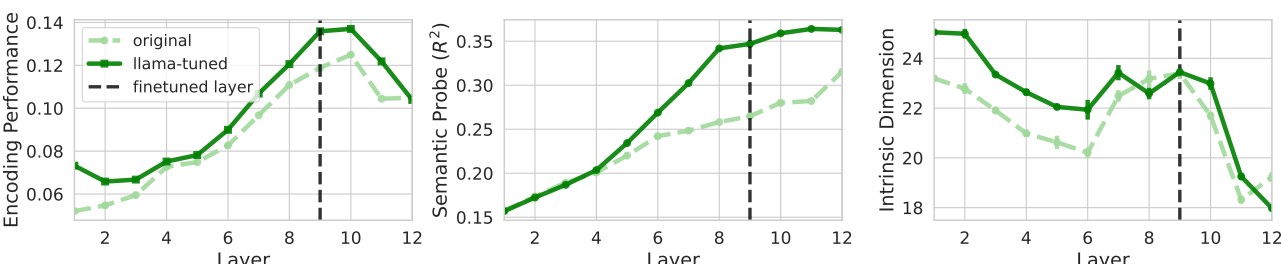

*Figure H.4.* **Finetuning using a different set of representations (Llama-3-8B increases encoding performance and semanticity, but not $I_d$.** To contrast with the braintuning experiments, we finetuned layer 9 of WavLM-base-plus—the same layer as the braintuning experiments—on the representations of the same input under Llama-3-8B. As a result, both encoding performance, shown here averaged across subjects, and semanticity improved in the middle layers, but not the $I_d$ (right).

# I. Random Fourier Feature Results for ECoG

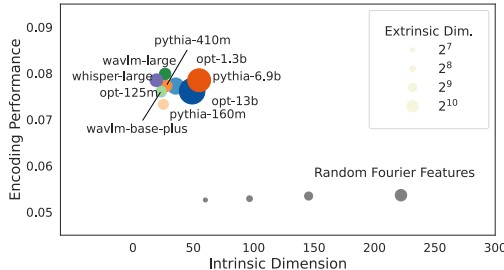

*Figure I.1.* **RFF Experiment, ECoG**. Companion to Figure 6. For the best LLM and speech layers (colors) as well as random Fourier feature spaces (gray) of increasing extrinsic dimension (○ size), we show the growth in $I_d$ (x-axis) with the corresponding growth in encoding performance averaged across ECoG subjects (y-axis).

# J. Distribution of Encoding Performance and its correlation to $I_d$ on the brain

*Table J.1.* **Correlation between $I_d$ and encoding performance in well-predicted voxels/electrodes.** In the well-predicted voxel/electrode subset ($R > 0.2$ for fMRI, $R > 0.1$ for ECoG), we show the mean $\rho(I_d, \text{encoding performance})$. The correlation is much higher for LLMs than for speech models, similar to the correlation on the non-filtered set of voxels/electrodes. There is a slight trend in which larger models show a stronger correlation between encoding performance and $\rho(I_d, \text{encoding performance})$.

|  | OPT | | | Pythia | | | WavLM | | Whisper |
|---|---|---|---|---|---|---|---|---|---|
|  | 125m | 1.3b | 13b | 160m | 410m | 6.9b | base | large | large |
| fMRI Subj. UTS01 | 0.57 | 0.65 | 0.68 | 0.20 | 0.65 | 0.61 | -0.32 | 0.46 | 0.75 |
| fMRI Subj. UTS02 | 0.65 | 0.70 | 0.71 | 0.33 | 0.59 | 0.57 | -0.06 | 0.41 | 0.86 |
| fMRI Subj. UTS03 | 0.71 | 0.73 | 0.75 | 0.28 | 0.65 | 0.66 | -0.08 | 0.49 | 0.85 |
| ECoG | 0.55 | 0.63 | 0.55 | 0.32 | 0.49 | 0.40 | 0.02 | 0.17 | 0.38 |

*Table J.2.* **In voxels and electrodes with higher encoding performance, encoding performance is better predicted by layerwise $I_d$ in LLMs, and less so in speech models**. The table shows $\rho(\text{EP}, \rho(I_d, \text{EP}))$, where EP is the encoding performance; the equivalent graph is the box-and-whisker plots in the appendix flatmaps. All correlations are significant with $p <$1e-2.

|  | OPT | | | Pythia | | | WavLM | | Whisper |
|---|---|---|---|---|---|---|---|---|---|
|  | 125m | 1.3b | 13b | 160m | 410m | 6.9b | base | large | large |
| fMRI Subj. UTS01 | 0.32 | 0.39 | 0.44 | 0.14 | 0.46 | 0.41 | -0.11 | 0.27 | 0.43 |
| fMRI Subj. UTS02 | 0.45 | 0.53 | 0.56 | 0.33 | 0.51 | 0.47 | -0.10 | 0.37 | 0.64 |
| fMRI Subj. UTS03 | 0.52 | 0.58 | 0.64 | 0.30 | 0.57 | 0.55 | -0.18 | 0.45 | 0.68 |
| ECoG | 0.36 | 0.56 | 0.44 | 0.38 | 0.45 | 0.39 | 0.07 | 0.07 | 0.25 |

### J.1. Encoding performance and $I_d$ flatmaps

Due to a large file size, remaining plots are found at the Github link: https://github.com/chengemily1/brain-id-abstract. In each plot, the first row respectively shows the encoding performances per-voxel and per-electrode for fMRI Subjects UTS01-3, and ECoG (redder is higher). The second row shows, for the same subjects, the Spearman correlations $\rho$ between the layerwise encoding performance and layerwise $I_d$ (redder is higher). It is evident, comparing the top and bottom rows, that the voxels and electrodes with higher encoding performance also have a stronger relationship between layerwise $I_d$ and layerwise encoding performance. This is seen by redder regions in the top row matching redder regions in the bottom row. In the bottom row, box-and-whisker plots show the growth in $\rho(\text{EP}, I_d)$, EP being encoding performance, is plotted with respect to encoding performance. This growth is consistent across subjects and imaging modalities for LLMs. For speech models, some of the best-predicted voxels are in the auditory cortex, which privileges low-level acoustic representation. For this reason, there is a inverse "U-shaped" trend in which the highest encoding performance voxels and electrodes may be less correlated with $I_d$ than voxels and electrodes that are slightly worse predicted.

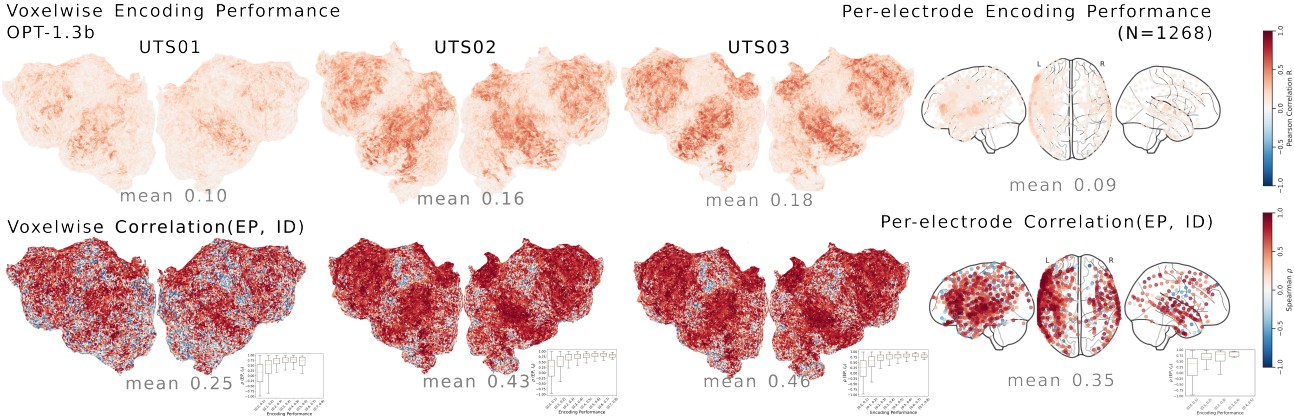

*Figure J.2.* **Encoding performance and $I_d$, OPT-1.3b.**

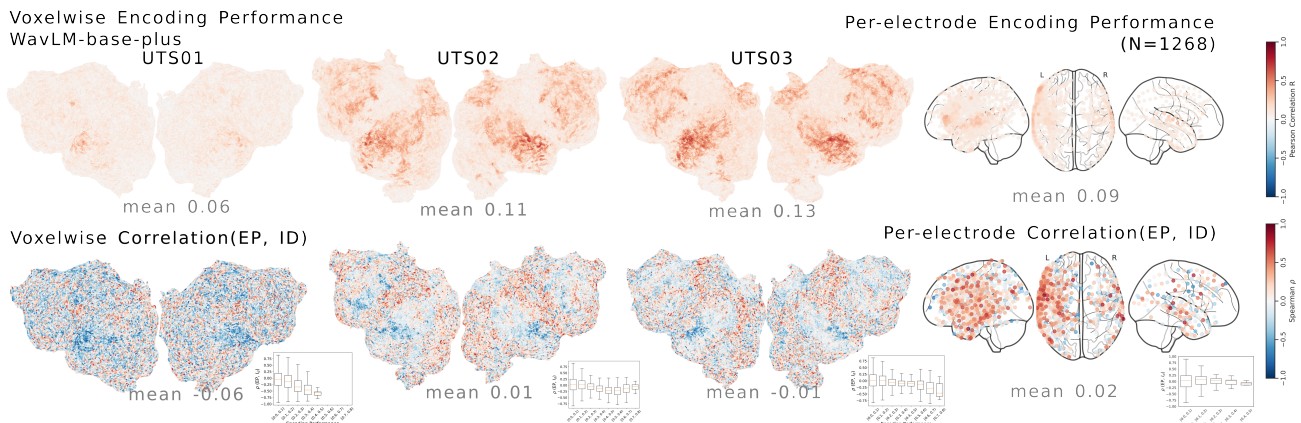

*Figure J.3.* **Encoding performance and $I_d$, WavLM-base-plus.**

