# OpenReview forum: "Abstraction Induces the Brain Alignment of Language and Speech Models"
_ICML.cc/2026/Conference — ICML 2026 regular_

### Official Review · Reviewer_7tis · 2026-03-01

**Soundness:** 3
**Presentation:** 3
**Significance:** 3
**Originality:** 3
**Overall Recommendation:** 4
**Confidence:** 4

**Summary:**

This paper investigates the reasons for brain-model alignment, specifically trying to understand why intermediate layers are the ones that obtain peak alignment. The authors suggest that the key driver of such alignment is “meaning abstraction”, rather than a shared predictive coding objective. Their main analyses are based on the correlation between brain encoding performance (linear encoding models for fMRI and ECoG) and the layerwise intrinsic dimension ($I_d$) of model representations, measured with the GRIDE estimator. The main claims are that: (1) across layers, the intrinsic dimension $I_d$ correlates with encoding performance and brain encoding performance peaks at $I_d$-peak layers; (2) $I_d$ and encoding performance dynamics over pretraining (Pythia checkpoints) are similar; and (3) “brain-tuning” a WavLM layer to improve fMRI encoding performance increases both $I_d$ and semantic content, suggesting a causal relationship between brain alignment, semantic content, and high-dimensional representations.

**Compliance With Llm Reviewing Policy:**

Affirmed.

**Final Justification:**

As explained in the acknowledgement, the additions made by the authors during the rebuttal address several of my methodological concerns.

I still believe that the current brain-tuning experiments should not be interpreted as strong causal evidence linking intrinsic dimension, semantic abstraction, and encoding performance, especially given that they are limited to a single model and modality. However, I acknowledge the computational constraints involved and appreciate the authors’ clarification and planned revisions to the discussion of limitations.

Overall, the paper addresses an important question about the representational basis of intermediate-layer brain–model alignment and introduces intrinsic dimension as a useful diagnostic for studying abstraction-based explanations of alignment. Despite remaining limitations regarding causal interpretation, I believe the analyses are of clear interest to the research community.

I am therefore increasing my soundness and significance scores to 3 (good), and my overall score to 4 (weak accept).

**Key Questions For Authors:**

1. Is the “best-of-128 lags” chosen on a validation split and then evaluated on a separate held-out test split, or is the same data used for both selecting the lag and reporting the correlation? If the latter, can you report a corrected estimate (nested selection / held-out evaluation)?
2. How stable are (1) the Id peak layer and (2) the correlation with encoding performance across a meaningful range of k values? Can you provide confidence bands over k rather than a single chosen k per model (especially given WavLM’s two-regime choice)?
3. In the brain-tuning experiment, do you just fine-tune layer 9 while the rest of the model is kept frozen? This seems to be the case at the moment. If not, please clarify it in the paper. If yes, why was this choice made?
4. Can you replicate the brain-tuning analysis on (1) a different WavLM layer, (2) Whisper, (3) ECoG modality, and (iii) an LLM? Without this, how can you prove that the mechanism generalizes beyond one tuned layer in one speech model?

**Limitations:**

Limitations are not currently adequately discussed. The paper should more directly acknowledge: (1) limited subject/stimulus diversity; (2) dependence/instability risks in $I_d$ estimation; (3) the limited experiments in support of causal evidence (brain-tuning analysis).

**Strengths And Weaknesses:**

## Soundness
### Strengths
- The paper includes experiments on both language and speech models, different model families (Pythia/OPT for language, WavLM/Whisper for speech), and brain modalities (fMRI, ECoG).

### Weaknesses
- The authors claim that “layerwise $I_d$ correlates highly to encoding performance globally across the models”. However, in the case of ECoG, the global correlation is 0.43, which is moderate rather than high. Additionally, the authors claim that “within each model, [...] the correlation is notably much higher than the alternative hypothesis (surprisal)”. However, Table 1 and Table H.1 show that for some of the models (OPT 1.3/13B and Whisper in Table 1, and OPT 1.3/13B, Pythia 140M, and Whisper in Table H.1) the correlation between encoding performance and surprisal is close to/as good as the correlation with $I_d$. The authors should discuss these results openly and say how they relate to the above-mentioned claim.

- It is not clear whether the results for ECoG modality are averaged across the 9 subjects. The authors should consider expanding also the analysis on the fMRI to more subjects, adding SEM across subjects in all tables and plots.

- The strongest causal claim is based on finetuning WavLM-base-plus through layer 9 on fMRI (just 2 subjects) and observing increased encoding performance, semantic probing performance, and $I_d$ (Figure 5; Appendix F). These experiments are not enough to demonstrate a “causal link” between semantic abstraction and encoding performance in general (esp. for LLMs). Additionally, it is not clear whether $I_d$ increases are a byproduct of optimization (e.g., changes in activation scale/geometry) or actual “semantic richness.” The authors should consider softening the causality claim, extending the brain-tuning experiments to LLMs and ECoG, add a baseline in which the models are simply fine-tuned (not brain-tuned) and check increases in $I_d$.

- The GRIDE estimator for $I_d$ requires the selection of the neighboring size k, which is selected by “visual inspection”, with one k per model (and even two regimes for WavLM early vs late layers). Given that one of the central claims in the paper is that $I_d$ peaks track brain peaks, then sensitivity to k should be analyzed, for instance showing correlation ranges over a band of k values. In Figure C.1, the selected k corresponds to the max $I_d$ of middle layers, while later layers (e.g. 22, 25, 28, 31) peak around $2^6, 2^7$. Why wasn’t a layer-specific choice of k made for all models?

- Across the results section, some central claims are based on experiments made on just 1 model and 1 brain modality (e.g. Pythia-6.9B for “$I_d$ and encoding performance dynamics over pretraining are similar”, and WavLM-base-plus + fMRI for “brain-tuning increases both $I_d$ and semantic content”). This does not support the generality of the claims that are made.

## Presentation
Overall, the paper is well-organized and understandable. However, some things remain unclear:
- Are the results shown for ECoG averaged across subjects? The authors should clearly say this in the captions.
- In Figure 2C, specify what the error bars refer to.
- In Figure 6, model labels are currently too mixed up. The authors should consider putting them in a legend outside the plot (e.g. on top).

## Significance
### Strengths
The paper addresses an important question in brain-model alignment research (its causes, why middle layers achieve peak alignment). Discovering a causal link between brain encoding performance and representation geometry (intrinsic dimension) could be impactful if robust, because it gives a simple diagnostic that might generalize beyond language to other modalities.

### Weaknesses
The practical impact is currently limited by methodological concern ($I_d$’s k choice) and by the limited causal evidence (brain-tuning analysis performed on just one model and brain modality).

## Originality
Most individual components in the paper (intermediate-layer peak alignment, intrinsic dimension peaks in deep networks, semantic probing, and representation changes during training) are well established in the literature. While the paper does not introduce a fundamentally new method or theory, the integration of these individual components is novel.

---

> ### Author Rebuttal · Authors · 2026-03-31
>
> Thanks for your valuable feedback. We respond point-by-point below. Figures that we reference below can be found [here](https://tinyurl.com/vs8svd7x).
>
> > for ECoG, the global correlation is 0.43, which is moderate rather than high...Table 1 and H.1 show that for some of the models the correlation… is close to/as good as the correlation with ID.
>
> We adjusted the claims in this section (e.g., “much higher than” -> “as good as or higher than”). To explicitly weigh how surprisal and ID contribute to encoding performance, we ran mixed effects models as suggested by Reviewer tNh6. This showed that ID explains away surprisal for fMRI and has a much larger effect size for both modalities (see Reviewer tNh6 response for details). This confirms the primacy of ID over surprisal in modulating encoding performance.
>
> > Is ECoG averaged across the 9 subjects?
>
> Yes; we will clarify in the final version.
>
> > consider expanding the fMRI analysis to more subjects, adding SEM in all tables and plots.
>
> We have added a third subject to the analysis, UTS01, to bring the work in alignment with other papers that have used this dataset in the past. The results are broadly consistent with the other subjects' results and do not affect our conclusions. We now report the mean over the fMRI subjects in the main paper ([see link](https://tinyurl.com/ycywvzcw)) with individual subjects in the Appendix. We will add the SEM to all relevant tables and plots in the final version.
>
> > Figure 2C error bars
>
> We revised the caption to “Pythia-6.9b’s layerwise Id ±1SD **(over 5 random data splits)** is shown over training”.
>
> > Is the “best-of-128 lags” …the same data used for both selecting the lag and reporting the correlation? If so, can you report a corrected estimate (nested selection / held-out evaluation)?
>
> We used the same test split for selecting the lag and reporting the correlation. We have revised to a cross-fold lag selection. There are almost no differences in previously-reported effects like layerwise or modelwise ordering as a result of this change, except in the earliest Pythia checkpoints where layerwise ordering differences between ECoG and fMRI were bigger. This change brings the two data modalities marginally closer to each other in observed behavior.
>
> > How stable are (1) the Id peak layer and (2) the correlation with encoding performance across a range of k values?
>
> Denti et al (GRIDE) suggest to select k where ID plateaus w.r.t. k. Following this suggestion, we chose $k$ such that it lies near critical points for all layers in the ID vs k plot. For simplicity, we chose one $k$ for all layers, as for all models it was possible to choose one k without considerably changing the ID curve. For WavLM (see [base-plus](https://tinyurl.com/ycx4h6xv) and [large](https://tinyurl.com/yyf7rf2w)), there was a phase shift about ⅓ of the way into the layers, so we chose one k for the first phase and another for the second, again for simplicity.
>
> These choices did not impact the shape of the ID curves, which are robust to k÷x 2 (see [figure](https://tinyurl.com/232n4jhy)). Trends in ID peak layers (see [table](https://tinyurl.com/tjy3yr94)) and correlation with EP ([table](https://tinyurl.com/y5fud2rb)) do not meaningfully change as a result. We added this analysis to the Appendix.
>
> > not clear if ID increases are a byproduct of optimization… or actual “semantic richness.” The authors should consider softening the causality claim
>
> We will soften the causality claim and add this point.
>
> > For brain-tuning, do you just fine-tune layer 9 while the rest of the model is kept frozen?
>
> Thanks for pointing this out. During brain-tuning, all layers through layer 9 are finetuned. We will clarify this in the final version.
>
> > Can you replicate the brain-tuning analysis on a different WavLM layer, Whisper, ECoG modality, and an LLM?
>
> Brain-tuning is computationally expensive and extending our results on more layers/models/modalities is unfortunately beyond our compute budget. We added another fMRI subject UTS01 as a middle ground and results generalize (semantic probes still running but see [EP/ID results](https://tinyurl.com/3wrh9x35)).
>
> > add a baseline in which the models are simply finetuned and check increases in ID.
>
> [Vattikonda et al. (2025)](https://tinyurl.com/5cx77utn)
>  showed that fine-tuning on non-brain data can increase EP for brain data. We replicated their setup, finetuning WavLM on LLaMa hidden states to check the ID. ID increased for most finetuned layers, staying the same for 2 of them ([Figure](https://tinyurl.com/5n75fhmt)). Probing results are running. For the final version we will explore finetuning on semantically "narrow" tasks, expecting ID to decrease.
>
> > acknowledge limited subject/stimulus diversity; dependence/instability risks in estimation; limited experiments in support of causal evidence
>
> Thanks for your suggestions. We have addressed these points in a new Limitations section in the Discussion.

---

> > ### Author Rebuttal · Reviewer_7tis · 2026-04-01
> >
> > Thank you for the detailed rebuttal. In particular, I appreciated:
> > 1. The addition of mixed-effects models to estimate the respective contributions of intrinsic dimension and surprisal to encoding performance.
> > 2. The adoption of cross-fold lag selection instead of selecting lags on the test split.
> > 3. The additional robustness analysis regarding the choice of the GRIDE neighborhood parameter k.
> >
> > I still believe that the current setup should not be interpreted as strong causal evidence for the relationship between intrinsic dimension, semantic abstraction, and encoding performance in the brain-tuning experiments, especially given that these analyses are limited to a single model and modality. However, I recognize the computational demands involved in extending these experiments.
> >
> > Overall, this work focuses on an important question, the representational basis of intermediate-layer brain–model alignment, and introduces $I_d$ as a tool to distinguish between abstraction-based explanations and predictive-objective accounts of alignment. Despite the remaining limitations regarding causal interpretation, which the author will acknowledge in a new Limitations section, I believe the analyses presented in the work are of clear interest to the research community and open several promising directions for future work.
> >
> > As a result, I am increasing my soundness and significance scores to 3 (good), and my overall score to 4 (weak accept).

---

> > > ### Author Response · Authors · 2026-04-01
> > >
> > > Thank you for the quick response and revision. We appreciate your feedback and it has helped improve the paper!

---

### Official Review · Reviewer_QADN · 2026-03-03

**Soundness:** 4
**Presentation:** 3
**Significance:** 3
**Originality:** 3
**Overall Recommendation:** 5
**Confidence:** 3

**Summary:**

The paper aims to understand the factors that drive good encoding scores between audio and text self-supervised models and brain activations. Specifically, why hidden representations at middle, rather than early or deeper, best predict brain activations. The paper considers brain recordings from both fMRI and ECoG modalities from subjects exposed to naturalistic auditory stimuli.

First, next-token surprisal and brain encoding score are compared across models and datasets. As expected the next-token surprisal decreases with deeper layers. In both fMRI and eCoG datasets, the highest encoding scores are achieved with moderate surprisal values, showing an inverse u-shape and thus, non-monotonic relationship.

Second, the intrinsic dimension of model representations is tested. The results show that in this case there exists a strong monotonic relation between encoding scores and intrinsic dimension which holds across models and recording modalities and language/audio models. Furthermore, both encoding scores and intrinsic dimensionality increase as a function of pretraining steps.
Third, the paper shows that those voxels/electrodes with higher encoding scores tend to also have a higher correlation between layerwise encoding scores and layerwise intrinsic dimensionality.

To confirm whether these findings are merely correlational, the authors fine-tuned layer 9 of the WavLM model to achieve higher fMRI encoding scores. This resulted in an increase intrinsic dimensionality.

Finally, the increase of encoding performance as a function of intrinsic dimensionality is shown not to be trivial by constructing random features with high intrinsic dimensionality. In this case, the encoding performance remains at chance, showing that intrinsic dimensionality alone is not enough to achieve high encoding scores.

**Compliance With Llm Reviewing Policy:**

Affirmed.

**Key Questions For Authors:**

Questions
- Can you say more about the voxels/electrodes where the correlation between intrinsic dimensionality and encoding score is high? Are these close to broca?
- Is it possible to show the correlations between linear dimension and encoding scores for different voxels/electrodes?
- How does intrinsic dimension relate to superposition? Can it be linked to any mechanistic finding in transformers?

**Limitations:**

yes

**Strengths And Weaknesses:**

Strengths
- The paper is clear and the experiments are very well executed and presented.
- The research question is important and shapes the understanding of the convergence between brains’ and AI models’ representations.
- Proposing intrinsic dimensionality as an important factor underlying encoding performance is a novel and potentially impactful idea.
- The paper is technically solid and the results are extensive and convincing.

Weaknesses
- Probing performance and intrinsic dimensionality do not fully agree.
- The emergence through pretraining is trivial, I would expect a similar result for the case of surprisal.
- Intrinsic dimension is a broad measurement, it would be important to link it mechanistically to Language/Audio models. It could be important to understand what exactly makes intrinsic dimension high. Understanding this could be crucial towards designing models with better abstracting capabilities.

---

> ### Author Rebuttal · Authors · 2026-03-30
>
> Thanks for your helpful feedback. We respond point by point below:
>
> > Probing performance and intrinsic dimensionality do not fully agree.
>
> Rather consistently across models, semantic probing performance rises and plateaus around the ID peak. In the larger LLMs, semantic probing performance plateaus exactly at the ID peak. For speech models the ID peak is slightly sooner, perhaps because mid-layers retain complexity from low-level acoustic features (but it is still the same neighborhood of layers). We replicated Cheng et al., 2025, where one can see that the ID peak corresponds to a phase of semantic decodability for an extended set of models. Similar to that paper, our observations are qualitative and about general trends, rather than making a claim about exact one-to-one correspondence between ID and probing performance.
>
>
> > The emergence through pretraining is trivial, I would expect a similar result for the case of surprisal.
>
> We believe the finding that ID increases over pretraining is somewhat surprising and nontrivial. A common narrative in the literature is that “learning amounts to compression”, which would suggest ID to start high for untrained models and decrease over training. Instead, the fact that it increases as the models learn sophisticated linguistic representations lends weight to the hypothesis that ID indeed tracks the building of a rich feature space.
>
> > Can you say more about the voxels/electrodes where the correlation between intrinsic dimensionality and encoding score is high? Are these close to broca?
>
> Broadly speaking, the correlation between the two metrics is high everywhere in the brain that processes language and semantics. However, we note that in regions which are predicted especially well, such as primary AC and Broca’s area, the correlation is even higher. We show this explicitly in Figure 4 of the paper.
>
> > Is it possible to show the correlations between linear dimension and encoding scores for different voxels/electrodes?
>
> We have added several examples of linear dimension / EP correlations for different voxels, along with the average correlation over all voxels, to the Appendix, see, e.g. [here](https://anonymous.4open.science/r/icml-rebuttal-24947-97E4/linear_dim_examples.png). In the plot, even rows correspond to linear dimension using the Participation Ratio, and odd rows to intrinsic dimension. Note that linear dimension correlated well to encoding performance for OPT (left column), but not for Pythia (right). Indeed, in preliminary experiments on LLMs, we found that PR and ID were correlated only for the OPT models. We believe this might be due to PR severely underestimating dimensionality in Pythia’s middle layers because of massive activations (Lee et al., 2025). This was a primary reason we used the intrinsic dimension for our analysis.
>
> > How does intrinsic dimension relate to superposition? Can it be linked to any mechanistic finding in transformers?
>
> Thanks for this question. The literature on ID in LLMs is still small, and a link hasn't been established between ID and degree of polysemanticity. We suspect that a high degree of polysemanticity/superposition might result in a space that looks like a union of manifolds, all possibly with different ID. This describes the intuition that different regions of the space might be described by different semantics, seen by subspaces of varying complexity. This is just a speculation but would be an interesting empirical question for future work.
>
> Linking ID to mechanistic findings in Transformers is very underexplored, perhaps because they concern different levels of granularity in the model. ID is a population-level measure, while mech interp tends to analyze, e.g., semantics of individual neurons. These two approaches address complementary questions, and many different neuron-level configurations could yield the same ID. The only work we know of that considers mech-interp and intrinsic dimension in the same paper is Shafran et al., 2026, who compute a local intrinsic dimension and then use locally linear subspaces for steering (but they don’t explore the relationship between ID and steering).

---

> > ### Author Rebuttal · Reviewer_QADN · 2026-04-03
> >
> > The authors correctly addressed my points. Thanks for the informative rebuttal.

---

### Official Review · Reviewer_tNh6 · 2026-03-12

**Soundness:** 3
**Presentation:** 4
**Significance:** 4
**Originality:** 4
**Overall Recommendation:** 5
**Confidence:** 3

**Summary:**

This paper addresses the question of what properties of LLM representations lead them to be highly predictive of human brain imaging data. Much prior work has suggested that the next-word prediction of LLMs is what drives LLM-brain alignment, but this paper instead presents evidence that it is the learning of a rich linguistic feature space that drives the alignment. To get at this point, the authors show that intrinsic dimensionality - a measure of how complex a feature space is - predicts brain encoding performance better than surprisal - a measure of next-word prediction performance. The authors also provide follow-up analyses showing that intrinsic dimensionality alone is not enough for strong brain encoding performance, which is evidence that it is in fact feature richness (which produces high intrinsic dimensionality as a downstream consequence) that drives strong LLM-brain alignment, rather than the intrinsic dimensionality being the driver itself.

**Compliance With Llm Reviewing Policy:**

Affirmed.

**Final Justification:**

I am retaining my high score because the rebuttal reinforced my strong prior assessment. My score is primarily based on the paper's strong soundness, originality, and significance, weighted equally with each other.

**Key Questions For Authors:**

Q1: Regarding W3 above, are there strong reasons to say that surprisal is not predictive of brain encoding performance? [This could affect my score slightly]

Q2: Not a question, but a comment: W1 and W2 are both big enough questions that it’s clearly beyond the scope of the current paper. However, making sure that these points are acknowledged could strengthen the paper (I believe that both are briefly gestured at, but addressing them more directly could be helpful).

Q3: There has been work showing that, as LLMs get better and better at next-word prediction, they predict human psycholinguistic measures better and better up to a point - but then they become *too* good at next-word prediction and start fitting humans less well (e.g., https://aclanthology.org/2023.tacl-1.20/). Where do the models studied here fit into this spectrum? The best fit seems to appear around 2 billion words of pretraining (https://aclanthology.org/2023.findings-emnlp.128/), so it would be helpful to know where models' training data falls (about 2 billion words? more? less?) to know how to interpret their surprisal predictivity. However, this question is unlikely to influence my score, so no problem if it's too much of a detour to look into this (especially since, to my knowledge, this "goldilocks" effect has only been shown for behavioral measures, not brain imaging measures).

**Limitations:**

yes

**Strengths And Weaknesses:**

Strengths:
S1 (significance, originality): The paper provides new evidence about a timely and important debate regarding LLM-brain alignment.

S2 (soundness): The experiments are thorough and extensive, covering six LLMs (and all layers within them) and two brain recording approaches, with similar results across conditions; this thoroughness supports the reliability of the findings. And the findings are clear (or at least as clear as brain-related work can be), showing a consistent trend of intrinsic dimensionality beating surprisal in terms of predictiveness.

S3 (soundness, originality): I really like the follow-up analyses that the authors did to (i) check for the causal nature of intrinsic dimensionality and (ii) to check whether intrinsic dimensionality is important in and of itself or if it is a downstream consequence of something else that is important. These analyses are useful for pinning down precisely what it is that the results show.

S4 (presentation): Although the paper is dense, it is clear, communicating all key details in an easy-to-understand way.

Weaknesses:
W1 (soundness): The paper leaves it a little unclear what it is about the intrinsic dimensionality that drives brain-LLM alignment. That is, the paper makes it clear that it’s not the intrinsic dimensionality itself that matters but rather some other factor that produces high intrinsic dimensionality, but it’s not clear what that other factor is - the paper suggests that it might be semantic richness (a claim supported by the causal analysis), but there is a potential concern that it’s instead something else that correlates with semantic richness. However, I don’t view this as a serious concern, as the paper does plenty of work already.

W2 (soundness): All models that are studied are next-word prediction models. This means that, in some sense, everything that they do arises from next-word prediction. E.g., to the extent that they derive rich representations, those representations are for the purpose of predicting the next word. The conclusions would be stronger if there had been models trained on some non-predictive task that had representations that were as rich and that were as predictive of the brain (which would of course be a huge task - I’m not saying that the authors need to do this, just making my point clearer). The authors do show that the layers that give the best surprisal are not the ones with the best encoding performance, which helps address this point, but I don’t think it is fully conclusive because even the layers that are less strong at surprisal might still be substantially influenced by the next-word prediction objective in ways that could not be explained purely by wanting rich representations.

W3 (soundness): The paper sometimes assumes a false dichotomy between next-word prediction vs. feature richness being the driver of LLM-brain alignment (e.g., saying “it is meaning abstraction, rather than prediction, that primarily drives the observed correspondence”, and saying “layerwise surprisal does not predict layerwise encoding performance”). But it seems plausible that both factors could be in play - indeed, e.g., in Table 1, surprisal and I_d both often give strong and statistically significant correlations. It could be illuminating to run a single statistical test (e.g., a mixed effects model) that includes both surprisal and I_d as predictors to suss out the extent to which each one is predictive and the extent to which each is predictive over and above the other - that might point to a narrative where both matter but I_d matters more, for instance.

One more note: It might be worth citing the following work, which makes some similar points about intrinsic dimensionality but for vision rather than language: Elmoznino and Bonner, “High-performing neural network models of visual cortex benefit from high latent dimensionality”

---

> ### Author Rebuttal · Authors · 2026-03-30
>
> Thank you for your time and helpful feedback. We respond point-by-point below:
> > All models studied are next-word prediction models. This means that, in some sense, everything that they do arises from next-word prediction...The conclusions would be stronger if there had been models trained on some non-predictive task that had representations that were as rich and that were as predictive of the brain.
>
> Note that while the LLMs are tasked on next-word prediction, WavLM and Whisper are not. WavLM is trained on a joint masked speech prediction and denoising objective, while Whisper is trained to transcribe the speech signal into text. Whisper does output text, but this is not the same as next-token prediction (predict the upcoming signal given context). We agree though that the field needs SOTA models that both predict the brain well and are trained on something other than next-token prediction.
> > (W3) even the layers that are less strong at surprisal might still be substantially influenced by the next-word prediction objective in ways that could not be explained purely by wanting rich representations.
>
> Thanks for this point. We acknowledged the impact of the training objective in the second to last paragraph of the conclusion: “In language models, these feature-rich layers do eventually serve next-token prediction”. To make this more explicit, we will add “An LLM or speech model’s training objective necessarily affects internal representations. Interestingly, all the speech-language models we tested achieve their training objective by passing through a phase of rich meaning abstraction (Cheng et al., 2025 and Jawahar, 2019); it is the layers in this phase, rather than the layers that best predict the next token, that are most aligned to the brain.”
> > (W3) The paper sometimes assumes a false dichotomy between next-word prediction vs. feature richness being the driver of LLM-brain alignment... it seems plausible that both factors could be in play. Indeed, e.g., in Table 1, surprisal and ID both often give strong and statistically significant correlations. It could be illuminating to run a single statistical test (e.g., a mixed effects model) that includes both surprisal and ID as predictors
>
> Thanks for this point. Surprisal gives correlations that are sometimes strong and significant, but it is generally not better compared to ID. We maintain that if surprisal were a true driver of encoding performance (EP), then we would expect high correlation to EP consistently across models (this is not the case, e.g. in Pythia). Instead, correlations are only high for models that incidentally happen to have an ID peak close to final layers, which suggests that surprisal correlations might be largely explained by ID.
>
> We ran mixed effects models for fMRI and ECoG: encoding_perf ~ surprisal + ID. Since surprisal and ID have different scales, we first z-scored them to compare effect sizes fairly.
> - For fMRI (averaged over subjects), surprisal was not significant, while ID was highly significant. Effect sizes for ID tended to be ~100x higher than surprisal. These results held for every subject, e.g., in UTS03 the effect sizes for ID and surprisal were 0.02 and 0.002 respectively, a 100x difference. This shows that every +1SD in ID tracks a 0.02 increase in EP, and every +1SD in surprisal tracks a 0.002 increase in EP.
> - For ECoG (avg over subjects), both regressors were significant, though ID was much more significant than surprisal (p-value=2.4e-56 vs. 4.8e-4). The effect size for ID (0.004) was 4x that of surprisal (0.001). These results also signal ID as being more important than surprisal in predicting EP.
>
> Overall, ID explains away surprisal for fMRI and has a much larger effect size across modalities. Thanks for suggesting the analysis, it is added to the Results.
>
> > W1 and W2 are both big enough questions that it’s clearly beyond the scope of the current paper. However, making sure that these points are acknowledged could strengthen the paper.
>
> Thanks for this point. We agree about W1 (let us know if anything remains unclear about W2). Representations that have high ID are very complex by definition and semantically rich (probing experiments, cf. Cheng et al., 2025). One implication of this is that these representations, being complex/rich, are good candidates for downstream tasks, in our case predicting the brain. Still, ID, like any dimensionality measure, is a coarse-grained dataset-level measure. It tells us how many degrees of freedom underlie a dataset without saying what the degrees of freedom are. We used probing to suggest that these degrees of freedom relate to higher-order linguistic features, but future work should tease apart what precise dimensions correspond to (dimension attribution is an open problem in the ID estimation literature). We’ll add this discussion to a Limitations section.
>
> > citing Elmoznino and Bonner paper
>
> Thanks! We are aware of this paper and we will cite it in the Discussion.

---

> > ### Author Rebuttal · Reviewer_tNh6 · 2026-04-03
> >
> > Thank you for the responses. I will retain my (high) score.

---

### Official Review · Reviewer_EbRy · 2026-03-13

**Soundness:** 3
**Presentation:** 3
**Significance:** 3
**Originality:** 2
**Overall Recommendation:** 4
**Confidence:** 5

**Summary:**

Recent linguistic brain encoding studies demonstrate that representations from language models can be used to accurately predict human brain activity when participants engage in text- or speech-evoked brain activity, suggesting parallels between language models and brain language representations. The authors of this paper mainly ask what the main reason is for the similarity between the two systems. Specifically, the authors ask whether the similarity is due to language models and the human brain processing representations similarly, or due to the intrinsic dimension of high-level abstractions present in the middle layers that result in higher brain predictivity. To address these questions, (i) the authors perform brain encoding to verify at which layers of language models the intrinsic dimension peaks and check the encoding performance at those layers, (ii) during language model pretraining, both intrinsic dimension and brain encoding performance grow in tandem, and (iii) finetuning language models with brain data further improves intrinsic dimension and semantic richness. Experimental results on two brain datasets, Moth Radio Hour and ECoG, reveal that higher intrinsic dimension with meaningful abstraction results in higher brain predictivity, but this is not due to surprisal effect, low-level features, or only high intrinsic dimension without meaningful abstraction

**Contributions:**

* This study provides new evidence on why the middle layers of language models show improved brain alignment compared to other layers, and reasons whether this effect is due to next-word prediction or surprisal, or due to higher intrinsic dimension with meaningful linguistic structure.
* Rigorous experiments: The evaluation of encoding performance vs. surprisal and vs. intrinsic dimension across layers shows meaningful insights across two brain datasets. The authors find that higher intrinsic dimension with meaningful structure is present more in the middle layers and results in better encoding performance, but this is not due to surprisal or simply higher intrinsic dimension with no meaningful structure.

**Compliance With Llm Reviewing Policy:**

Affirmed.

**Final Justification:**

I have read the authors' replies and believe they have partially addressed my concerns. Overall my concerns of interpretability of intrinsic dimension, the extent of novelty beyond prior work, and the incomplete disentanglement of model-specific factors are only partially resolved.

Therefore I am raising significance score as 3 and overall score as 4.

**Key Questions For Authors:**

**Major Comments/Questions:**

* Architectural differences and model scale: In this paper, the authors use the two different language and speech models. Since these evaluated models differ in architecture, pretraining data, and scale, how much do these factors influence the observed relationship between intrinsic dimension and encoding performance across layers?
* What is the contribution of surface and syntactic features in early and middle layers?: Prior studies [Jawahar et al. 2019] have shown that language model layers encode linguistic information hierarchically from early to later layers. In the context of intrinsic dimension across layers, did the authors control for both surface and syntactic features? More importantly, under the geometric interpretation, how much do these two types of features contribute to the observed intrinsic dimension?
* Limited subjects from Moth-Radio-Hour: Since the Moth Radio Hour dataset includes 6 subjects, it is unclear why the authors conduct the analysis on only 2 subjects. I therefore recommend that the authors extend the analysis to all participants.

**Minor Comment:**
* Line 117: bullet point is 4 not 3.

**Limitations:**

Yes

**Strengths And Weaknesses:**

**Strengths:**

I found this work to have the following strengths:

* **Soundness:** The use of higher-order intrinsic dimension to explain why middle layers show improved brain alignment is methodologically sound, and the authors provide evidence on both text and speech language models. The model choices, such as the OPT and Pythia series for text and speech models such as WavLM and Whisper, are appropriate for this analysis. Higher intrinsic dimension with meaningful content shows a peak in the middle layers and results in better encoding performance in those layers.  The experimental protocol across two brain modalities also seems appropriate, as the authors compare brain encoding performance on each brain dataset and analyze the factors that result in better encoding performance. Overall, the approach is methodologically sound, and supported by experiments across both text and speech language models.

* **Presentation:** The motivation for understanding why language model representations better predict brain activity, through two interesting questions such as (i) whether this is due to similar representations or (ii) whether this is because they largely share meaningful abstraction in features, is clearly presented in the introduction. The authors present evidence through different experiments, such as middle layers exhibiting higher encoding performance due to higher intrinsic dimension with meaningful abstraction, language model layers with the best next-word prediction (i.e., surprisal) not predicting the best encoding performance, intrinsic dimension peaking at certain layers of language models and those layers resulting in better encoding performance, and increased encoding performance with increasing semantic content and intrinsic dimension, which clearly presents the findings. Overall, the methodology is clearly presented. In addition, the experiments on the two brain datasets covering two modalities (fMRI and ECoG) are well described, including the experimental setup and comparisons with prior work, and provide new evidence for the similarity between language models and the human brain’s language comprehension.

* **Originality:** Although several studies explore the reasons behind the similarity of language model representations in the middle layers that result in better encoding performance [Oota et al. 2023], this study further expands this analysis by measuring intrinsic dimension across the layers of language models and comparing it with encoding performance, which is methodologically novel. Although certain evaluations such as linear probing [Oota et al. 2023] and surprisal effects [Schrimpf et al. 2021; Caucheteux et al. 2021] have been studied before, this study observes new findings that the best layer for next-word prediction does not predict brain encoding, suggesting that these findings are novel. Empirically, the authors provide new evidence that language models show better encoding performance because the middle layers consist of higher intrinsic dimension with shared abstraction.

[Oota et al. 2023] Joint processing of linguistic properties in brains and language models, NeurIPS-2023

[Schrimpf et al. 2021] The neural architecture of language: Integrative modeling converges on predictive processing, PNAS

[Caucheteux et al. 2022] Brains and algorithms partially converge in natural language processing. Communications Bilogy

[Caucheteux et al. 2023] Evidence of a predictive coding hierarchy in the human brain listening to speech. Nature Human Behavior

* **Significance:** This work is significant in that it contributes to understanding the similarity in language comprehension processing between AI systems and human brain recordings. In particular, this study helps provide new evidence that the best layer next-word prediction does not result in better encoding performance. Similarly, higher intrinsic dimension with no meaningful structure results in lower encoding performance. Overall, the findings are significant across two brain datasets, showing that higher intrinsic dimension with meaningful content results in better encoding performance at middle layers.

**Weaknesses:**

* **Limited novelty in the use of intrinsic dimension:**  The use of intrinsic dimension to better understand layerwise abstraction is already explored in prior work by Cheng et al. (2025), which showed that intrinsic dimension peaks in middle layers and reflects an abstraction phase in transformer language models. This paper extends this idea to brain encoding and reports related findings in the context of fMRI and ECoG alignment. Therefore, the novelty lies more in the brain-alignment extension than in the intrinsic-dimension hypothesis itself. In addition, some evaluations in this paper, such as linear probing and surprisal analyses, are conceptually connected to prior work, although the brain-encoding and brain-finetuning experiments are new with exploration of intrinsic dimension. Finally, the claim that middle layers always provide the best encoding performance may be too strong, as the best layer encoding performance can depend on the architecture, dataset, and benchmark, with some works reporting middle layers [Toneva et al. 2019, Oota et al. 2023] and others middle-to-late layers [Antonello et al. 2023] as most predictive of brain activity. I recommend that the authors provide a stronger explanation for why some studies report middle layers while others report middle-to-late layers as best for encoding performance, and clarify how the intrinsic-dimension findings relate to these cases.

[Antonello et al. 2023] Scaling laws for language encoding models in fMRI, NeurIPS-2023

* **Limited evaluation on intrinsic dimension:** The authors claim that intrinsic dimension peaks in the middle layers and is associated with higher encoding performance. However, it remains unclear what kinds of linguistic information contribute to this intrinsic dimension, especially given that the paper also includes probing analyses. In particular, if only a higher intrinsic dimension with meaningful structure leads to better encoding performance, the paper should explain more clearly what this shared abstract structure consists of. For example, does it mainly reflect discourse-level, morphological, semantic, or syntactic information?  Without such analysis, the interpretation remains vague: intrinsic dimension is repeatedly called as if it were explanatory, but the paper does not clearly specify what linguistic content these dimensions correspond to. This makes the claim of "meaningful abstraction" feel underspecified and weakens the interpretability of the main finding.

* **Architectural differences and model scale:** The authors considered two different language models, OPT and Pythia, and two speech language models, WavLM and Whisper. However, these models differ in architecture, pretraining data, and scale. As a result, it is unclear whether the relationship between intrinsic dimension and encoding performance reflects a general property of brain alignment or is partly controlled by model-specific factors.  If intrinsic dimension correlates with encoding performance,the authors should analyze whether architecture systematically influences intrinsic dimension across layers. In addition, how similar are the geometric representations across different architectures, especially at the best encoding layers, and do those layers exhibit similar linguistic properties?

---

> ### Author Rebuttal · Authors · 2026-03-30
>
> Thank you for your time and helpful feedback. We respond point-by-point below.
> >  some works report middle layers [Toneva et al. 2019, Oota et al. 2023] and others middle-to-late layers [Antonello et al. 2023] as most predictive of brain activity. I recommend the authors provide a stronger explanation for why some studies report middle layers while others report middle-to-late layers as best for encoding performance, and clarify how the ID findings relate to these cases.
>
> The claims presented in the cited papers are not incompatible with each other nor with our results. Differences in where the encoding performance (EP) and ID peak is in the model can be largely attributed to differences in training duration and scale. Our paper directly shows this in the Pythia results (Figs. 2A, H.1), which shows that the EP/ID peak moves farther back both in later training epochs and for larger models. This movement within the same model family suggests the exact location of the EP/ID peak is largely dependent on details of the training regime of the model. The Antonello et al. paper that the reviewer cited also indirectly demonstrates this in the differences between the OPT and LLaMa models, where the better trained but otherwise architecturally identical LLaMa has a much earlier encoding performance peak than its comparatively sized OPT analogue.
>
> >  It remains unclear what kinds of linguistic information contribute to ID… In the context of ID across layers, did the authors control for both surface and syntactic features? More importantly, under the geometric interpretation, how much do these two types of features contribute to the observed ID?
>
> Thanks for this comment. The best tools to do this are the ones we have used (probes, Fig 3), where experiments showed early layers to index surface-level features and deep layers to index abstract features (cf Cheng et al., Skean et al., Lad et al., 2025). This supports interpreting the ID peak as a signature of linguistic abstraction and the construction of a complex feature space. We emphasize that ID is a general measure of representational complexity and does not tell us what the dimensions are per-se (thus one cannot control it for X or Y feature). Precise mechanistic attribution of intrinsic dimensions to interpretable features is an open question in the ID literature. We added this discussion to the Limitations.
>
> > models differ in architecture, pretraining data, and scale...how much do these factors influence the observed relationship between ID and encoding performance across layers?
>
> We considered 3 **sizes** from OPT, 3 from Pythia, 1 from Whisper and 2 from WavLM. Fig. 2 shows that both encoding performance and ID generally increase with scale within the same family.
>
> Fine-grained **pretraining data** attribution to the EP/ID relationship is an important direction for future work, but is outside the scope of this project. In general the training sets for these models are too big to analyze effectively, as well as not publicly available (e.g. for OPT).
>
> However, we did analyze the impact of pre-training (Fig. 2 bottom, Pythia 6.9b over 8 checkpoints). Fig 2 shows that EP/ID both increased with pretraining, tying together existing results from the literature (Cheng et al. and Alkhamissi et al., 2025). Additional training appears to move the abstractive peak to earlier in the model, as mentioned in our earlier response.
>
> Attributing **architectural** choices to the ID-EP relationship is also very difficult. Architecture reflects the choices of the organizations that train models, so it is extremely hard to disentangle all these factors without pretraining LLMs ourselves with a much wider search space. The space of possible architectural choices is effectively unbounded and difficult to systemically search. Unfortunately, we do not have the computational means to compellingly perform this analysis, but it is an interesting direction for future work by better-resourced colleagues.
>
> > Including more fMRI participants
>
> While there are 6 subjects in the fMRI dataset, we limited our analysis to the 2 subjects with the highest data quality primarily due to very heavy computational constraints, as thousands of node-hours of compute have been expended to provide the analyses currently in the paper. 3 remaining fMRI subjects have only 5 sessions of data due to lower data quality and are not well suited for these analyses. We have now added results for UTS01 (the final subject with ~20 sessions of data) to extend the analysis to the same set of subjects in the Antonello et al. paper which the reviewer cites in another comment. We find that this addition does not affect any of our conclusions. We will update the manuscript in the final version to put the fMRI results averaged over subjects in the main text and the per-subject breakdown in the appendix. [Link to the updated Table 1](https://anonymous.4open.science/r/icml-rebuttal-24947-97E4/average_over_participants_table1.png)

---

> > ### Author Rebuttal · Reviewer_EbRy · 2026-04-02
> >
> > Fully addressed:
> > * The variability of the encoding performance peak at layers appears to depend on the model's training regime and scale.
> > * The authors extended the analysis to an additional fMRI subject.
> > * Architecture control is only partially addressed, but I agree with author's clarification given the scope of the work.
> >
> > Partially addressed:
> >
> > * Interpretation of intrinsic dimension: Although the authors argue that probing is a possible way to understand differences between early and deeper layers, prior studies such as Oota et al. (2023) and He et al. (2023) already used probing-task analyses to provide this kind of interpretation.
> > * The authors state that intrinsic dimension is a general measure of representational complexity, and I agree that ID by itself does not reveal what the dimensions correspond to. However, the rebuttal does not clearly justify why the probing results should be taken as evidence for the nature of the intrinsic dimensions.
> > * My first weakness still stands: layerwise abstraction has already been explored in prior work, such as Cheng et al. (2025). The main contribution of this paper is therefore the extension of this analysis to fMRI and ECoG experiments. While I agree that the empirical findings are interesting, the core conceptual questions remain largely unchanged.
> > * The authors reasonably note that fine-grained pretraining data attribution is out of scope, but my concern about disentangling model-specific factors remains largely unresolved.
> >
> > Overall, the rebuttal improves clarity on the certain questions, but main conceptual weakness remains: the paper still does not clearly explain what "meaningful abstraction" actually consists of.

---

> > > ### Author Response · Authors · 2026-04-02
> > >
> > > > Although the authors argue that probing is a possible way to understand differences between early and deeper layers, prior studies such as Oota et al. (2023) and He et al. (2023) already used probing-task analyses to provide this kind of interpretation.
> > >
> > > We agree that other works have also used probing to interpret the evolution of layer function. Probing is now quite standard in the field, and we do not claim to be the first to use it. Instead, we view our probing analyses as supplementary to our claim that ID is a measure of general abstraction: we used probing to establish that high ID corresponded to a phase of “deeper” linguistic processing for each model we tested. This was in part in order to ensure the results from Cheng et al. (relating an ID peak to higher-order processing) generalized to the specific models we were using. It is not the main contribution of the paper but is simply meant to buttress our claims about the ID metric.
> > >
> > > > The authors state that intrinsic dimension is a general measure of representational complexity, and I agree that ID by itself does not reveal what the dimensions correspond to. However, the rebuttal does not clearly justify why the probing results should be taken as evidence for the nature of the intrinsic dimensions.
> > >
> > > Thanks for allowing us to clarify. This paper does not investigate what particular intrinsic dimensions correspond to. Instead, we are interested in the ID insofar as it fluctuates (i.e., is high or low) over model layers as a *general* measure of representational complexity. Cheng et al. established that high ID is a correlative marker of the availability of higher-order linguistic information in a representation space. We did the same here as part of our analysis, extending their experiments to more LLMs and speech-audio models.
> > >
> > > There are theoretical reasons for which expansion in the ID should index rich/abstract processing. Dimensionality expansion has been classically associated with rich feature spaces in machine learning (the “kernel trick”, [Ansuini et al., 2019](https://arxiv.org/abs/1905.12784)) and theoretical neuroscience (see, e.g., [Fusi et al., 2016](https://pubmed.ncbi.nlm.nih.gov/26851755/), [Rigotti et al., 2013](https://www.nature.com/articles/nature12160)). Indeed, dimensionality expansion via nonlinear transformations of the input is a well-known mechanism by which complex nonlinear features get disentangled, such that they can be decoded with a linear readout. In our case, linear readout, operationalized with linear probing experiments, showed that these features are syntactic/semantic in the middle layers corresponding to high ID. Thus, both ID and probing reveal simultaneous properties of the *representation space*, our object of interest; we do not seek to directly relate probing to particular intrinsic dimensions.
> > >
> > > > My first weakness still stands: layerwise abstraction has already been explored in prior work, such as Cheng et al. (2025). The main contribution of this paper is therefore the extension of this analysis to fMRI and ECoG experiments. While I agree that the empirical findings are interesting, the core conceptual questions remain largely unchanged.
> > >
> > > We agree with the reviewer that the primary novel contribution of this work is not an analysis of layerwise abstraction independent of the brain, which as the reviewer correctly notes has existed in the past and, as evidenced by our citations, is foundational for this paper. Instead, we see this manuscript as connecting this prior body of work on layerwise abstraction to the open, highly-debated, active research question of what it is about the representations of speech and language models that induces their alignment with the brain. We consider the bridge that this work builds between manifold theory and computational language neuroscience as an important and novel one, even if past work has independently analyzed representational manifolds in the more constrained context of ANNs such as in Cheng et al. As a result of this, we agree that, at its core, our work will likely be more interesting to neuroscientists than theoreticians. We hope the reviewer will reassess their characterization of the novelty of our work as merely an “extension”, given this justification and the broad agreement among the other reviewers about its novelty.
> > >
> > > > my concern about disentangling model-specific factors remains largely unresolved.
> > >
> > > The current iteration of our manuscript analyzes such model-specific factors as scale, training time and modality (speech vs. language). While there are other factors that we could conceivably consider evaluating (the existence of residual connections, Transformer vs. LSTM, hidden state size, choice of nonlinearity, etc.) we are unsure whether these analyses would add any additional context to the underlying question of the paper, which is what is it that induces the linear relationship between representations from neural networks and the brain.

---

### Decision · Program_Chairs · 2026-04-30

**Decision:**

Accept (regular)

**Comment:**

This work investigates the reasons for alignment between language & speech models and human brain activity. Specifically, it examines whether strong brain alignment is due to next-token prediction or to the emergence of semantically rich representations in intermediate model layers. The reviewers highlight the timeliness of the research question and the comprehensive empirical evaluation, spanning multiple language and speech models, model scales, and two brain recording modalities. The paper can be strengthened by including a discussion on why intrinsic dimensionality tracks semantic abstraction, and a hypothesized relationship between intrinsic dimensionality and known linguistic or mechanistic factors in the model. Overall, the strong and comprehensive empirical evaluation make this work a good contribution to the emerging NeuroAI area.